# Estimation of Treatment Effects in Extreme and Unobserved Data

**Jiyuan Tan**[*]
Department of Management Science and Engineering
Stanford University
jiyuantan@stanford.edu

**Jose Blanchet** [†]
Department of Management Science and Engineering
Stanford University
jose.blanchet@stanford.edu

**Vasilis Syrgkanis**[‡]
Department of Management Science and Engineering
Stanford University
vsyrgk@stanford.edu

## Abstract

Causal effect estimation seeks to determine the impact of an intervention from observational data. However, the existing causal inference literature primarily addresses treatment effects on frequently occurring events. But what if we are interested in estimating the effects of a policy intervention whose benefits, while potentially important, can only be observed and measured in rare yet impactful events, such as extreme climate events? The standard causal inference methodology is not designed for this type of inference since the events of interest may be scarce in the observed data and some degree of extrapolation is necessary. Extreme Value Theory (EVT) provides methodologies for analyzing statistical phenomena in such extreme regimes. We introduce a novel framework for assessing treatment effects in extreme data to capture the causal effect at the occurrence of rare events of interest. In particular, we employ the theory of multivariate regular variation to model extremities. We develop a consistent estimator for extreme treatment effects and present a rigorous non-asymptotic analysis of its performance. We illustrate the performance of our estimator using both synthetic and semi-synthetic data.

## 1 Introduction

We are interested in studying the effect of treatment e.g., different policies and drugs, on rare yet impactful events such as large wildfires, hurricanes, tsunamis and climate change. These kinds of events happen at an extremely low frequency, but they can cause considerable damage to properties and pose serious threats to people's lives. For instance, we may want to know the effect of more

---

[*]Jiyuan Tan was partially supported by NSF Award IIS-2337916.

[†]Jose Blanchet gratefully acknowledges support from DoD through the grants Air Force Office of Scientific Research under award number FA9550-20-1-0397 and ONR 1398311, also support from NSF via grants 2229012, 2312204, 2403007 is gratefully acknowledged.

[‡]Vasilis Syrgkanis was supported by NSF Award IIS-2337916.

39th Conference on Neural Information Processing Systems (NeurIPS 2025).

infrastructure investment or other kinds of precautionary policies on earthquakes. In many applications – from financial risk to environmental policy – it isn't enough to know how a treatment changes the average outcome; decision-makers care about whether it alters the extreme tail. More formally, we may want to estimate the effect of treatment $D$ on outcome $Y$, conditioning on some extreme events. Estimating this kind of effect can help policymakers evaluate the impact of a policy and choose the best policy to reduce economic loss and save more lives when disasters happen.

Despite its clear importance, existing methods fall into two largely disconnected strands, each of which cannot fully address this question. One approach comes from the causal inference literature. Causal inference provides a comprehensive framework for counterfactual reasoning. Causal effect estimation is an important problem in this area, which finds wide applications in healthcare, education, business decision-making, and policy evaluation. Classic causal inference literature mainly focuses on estimating the average effects among certain groups. Little attention is paid to the causal effect on rare events. The scarcity of extreme data makes inference more challenging than in classic settings. As a result, naively applying classic causal inference estimation methods will produce poor results with large statistical error. For example, when making policies about earthquakes, we are usually unable to see a strong signal from historical data, as large earthquakes rarely occur and there are fewer samples in the dataset.

On the other hand, the Extreme Value Theory (EVT) studies the tail behaviors for statistical distributions, which provides the ideal tools for analyzing rare events. However, this approach does not take the data structure into consideration. In particular, it does not accommodate counterfactual treatments or adjust for covariates, so it cannot tell us what would happen under an intervention.

To bridge these gaps, we combine causal inference theory with EVT to provide a novel framework for extreme effect measurement. Following researches in EVT Coles et al. [2001], we use a multivariate regularly varying variable $U$ to model extreme events. The rare event can be modeled by the event $\{\|U\| > t\}$ for large $t$. Our proposed estimand can be viewed as the Average Treatment Effect (ATE) conditioning on $\{\|U\| > t\}$ with rescaling as $t$ increases to infinity. Detailed definition and explanation can be found in Section 3. Estimation is challenging because the limiting tail distribution is unknown and must be inferred from finite samples. To improve data efficiency and inference accuracy, we combine tail observations with moderate-frequency data in an extrapolation scheme, leveraging EVT insights alongside causal-inference techniques to achieve efficient estimation.

To the best of our knowledge, we are not aware of any work in the literature that considers this problem. In this paper, we take the first step to measure the treatment effect on extreme events. To be more specific, our contributions can be summarized as follows.

1. We propose a measure for the treatment effect on rare events named Normalized Extreme Treatment Effect (NETE), which essentially measures the magnitude of treatment on tailed events.

2. We develop two consistent estimators for NETE—a doubly robust (DR) estimator and an inverse propensity weighting (IPW) estimator—by combining recent advances in multivariate tail–dependence estimation Zhang et al. [2023] with double machine learning methodology Chernozhukov et al. [2018], and derive finite-sample, non-asymptotic error bounds.

3. Synthetic and semi-synthetic experiments demonstrate a good practical performance of our proposed estimator as compared to baseline estimators adapted from standard causal inference literature.

**Related Work**  We briefly review some relevant literature in EVT and causal inference. Coles et al. [2001] provides a comprehensive introduction to EVT. A large amount of work focuses on the univariate setting Davison and Smith [1990], Leadbetter [1991], Pickands III [1975], Smith [1989]. Recently, there have been many recent works on the multivariate generalization of these results Avella-Medina et al. [2022], Zhang et al. [2023]. Causal effect estimation is a classical problem in causal inference [Rubin, 1974]. Common estimators include IPW [Rosenbaum and Rubin, 1983], DR methods [Bang and Robins, 2005, Kang and Schafer, 2007, Chernozhukov et al., 2016, 2017, 2018], Targeted Maximum Likelihood Estimation (TMLE) [van der Laan and Rubin, 2006]. There have been some efforts in the literature trying to combine the two research areas. Gissibl and Klüppelberg [2018] considers a special kind of Structural Causal Model (SCM) and shows that the proposed SCM is a kind of max-linear model. They also analyze the asymptotic distribution of their model. Chernozhukov and Du [2006], Zhang [2018], Deuber et al. [2024] consider the task of estimating the

extreme Quantile Treatment Effect (QTE). Aloui et al. [2023], Bodik [2024] assume the outcome falls into the domain of attraction and define the extreme treatment effect as the difference between EVI. Huang et al. [2024] study the estimation of extreme quantile effect and extreme average treatment effect in the continuous treatment setting. The other line of work Gnecco et al. [2021], Mhalla et al. [2020], Bodik et al. [2023] uses EVT to help causal discovery. However, we want to point out that the problems these works consider are quite different from our setting. The most similar setting would be extreme QTE estimation Chernozhukov and Du [2006], Zhang [2018], Deuber et al. [2024], but the QTE still cannot capture on how the expectation of the outcome changes under intervention. While all the previous work models extreme events using univariate variables, we model it using multivariate regularly varying variables. Moreover, our work can also be placed in the broader context in the literature that studies causal effects of treatment beyond the mean, for example, distributional effect Abadie [2002], Gautier and Hoderlein [2011], Hohberg et al. [2020], Ratio- and Log-Ratio-Based Effects Cole and Hernán [2002], VanderWeele [2013], CATE Shalit et al. [2017], Wager and Athey [2018], Abrevaya et al. [2015].

## 2 Preliminary

**Causal Inference.** We use the potential outcome framework Rubin [1974] in this paper. Let $W, D, Y$ be the covariate, binary treatment and outcome, respectively. We denote $Y(d)$ to be the potential outcome when the treatment is set to be $d$ and assume consistency i.e., $Y(D) = Y$ throughout the paper. The Average Treatment Effect (ATE) is defined as

$$\text{ATE} = \mathbb{E}[Y(1) - Y(0)].$$

The ATE measures the effect of a treatment on the outcome $Y$. In the policy-making example, $D$ is an indicator of whether to use the policy or not. $W$ is a covariate that may influence $D$, like the geographic features of a place, which will influence the local government's decision on policies, and $Y$ can be the economic loss. The ATE in this case provides information about how much loss can be saved if a policy is enforced. Under the following exogeneity and overlap assumptions, the ATE can be identified using the g-formula $\mathbb{E}[\mathbb{E}[Y|W, D = 1] - \mathbb{E}[Y|W, D = 0]]$.

**Assumption 2.1** (Exogeneity). The data generation process satisfies $(Y(1), Y(0)) \perp D \mid W$.

Besides, the following overlap assumption is also often needed for non-asymptotic analysis.

**Assumption 2.2** (Overlap). There exists constant $c_p \in (0, 1/2)$ such that the propensity score $p(w) = P(D = 1 \mid W = w) \in [c_p, 1 - c_p], \forall w \in \mathcal{W}$.

This assumption ensures that there is no extremely high or low propensity, which can make estimators unstable.

**Extreme Value Theory.** The study of extremity is mainly concerned with the tail behaviors of heavy-tailed distributions, which are often modeled by the regularly varying distributions. In this paper, we modeled extremity by multivariate regularly varying distributions.

**Definition 2.3.** A random variable $U \in \mathbb{R}_+^d$ is called regularly varying with index $\beta \in (0, \infty)$ if for any norm $\| \cdot \|$ in $\mathbb{R}^d$ and positive unit sphere $\mathbb{S}^+ = \{x \in \mathbb{R}_+^d : \|x\| = 1\}$, there exists a probability measure $S(\cdot)$ on $\mathbb{S}^+$ and a sequence $b_n \to \infty$ such that $n \, P((\|U\|/b_n, U/\|U\|) \in \cdot) \xrightarrow{w} c_U \cdot \nu_\beta \times S$ for some constant $c_U > 0$, where $\cdot \times \cdot$ is the product measure and $\nu_\beta([r, \infty)) = r^{-\beta}$ for all $r > 0$.

The parameter $\gamma = 1/\beta$ is called the Extreme Value Index (EVI), which characterizes the decay rate of the tail. Notice that this definition implies that as $b_n \to \infty$, the norm of and $\|U\|$ and its angle $U/\|U\|$ become asymptotically independent. We will leverage this fact for estimation in later sections. A typical example of regularly varying distributions is the Pareto distribution.

**Definition 2.4.** The density of a Pareto (type II) distribution with index $\beta \in (0, \infty)$ is $f(x) = \beta(1 + x)^{-(\beta+1)}, \forall x > 0$.

Definition 2.3 implies that the rescaled norm of a regularly varying variable is asymptotically a Pareto distribution.

**Notations.** In the rest of the paper, we use $\| \cdot \|$ and $\| \cdot \|_1$ as a shorthand for $\ell_1$-norm. We use the asymptotic order notation $o(\cdot), O(\cdot)$ and $\Theta(\cdot)$. We use $\mathbb{E}[\cdot]$ to represent expectation. For a matrix

$A$, we denote $A_{\cdot,i}$ to be its $i$-th column. $\text{Unif}([a,b])$ is the uniform distribution on interval $[a,b]$ and $\text{Ber}(p)$ is the Bernoulli distribution with expectation $p$. $\perp$ represents the independence relationship between two random variables.

# 3 Treatment Effect on Extreme Events

## 3.1 Extreme Semi-parametric Inference

While standard causal estimands capture average effects of $D$ on $Y$, they obscure what happens in the tails—i.e., when rare, high-impact events occur. To address this, we model rare events with an explicit extreme factor $U$. The data we consider is of the form $\{(X_i, D_i, Y_i, U_i)\}_{i=1}^N$, where $D$, and $Y$ are as defined in Section 2, $W = (X, U)$ is the covariate, $U$ is the extreme part of the covariate. We use $\|U\|$ to model the severity of rare events—large norms indicate more extreme realizations. For example, in a hurricane-loss application, $U$ might be the vector of maximum wind speed, rainfall, and storm surge; $X$ the region's location; $D$ the level of infrastructure investment; and $Y$ the resulting economic loss.

In what follows, we introduce a novel estimand that quantifies the causal effect of $D$ on $Y$ specifically in the tail region defined by large $\|U\|$. We then establish conditions for its identification under multivariate regular variation and propose two consistent estimators. We will make the following i.i.d. assumption.

**Assumption 3.1.** The random variables $\{(X_i, D_i, Y_i, U_i)\}_{i=1}^N$ are i.i.d.. Furthermore, $U$ is regularly varying.

We are interested in the effect of treatment on the tail events of $U$. Similar to ATE, a naive definition of the extreme treatment effect would be

$$\theta^{\text{ETE}} = \lim_{t \to \infty} \mathbb{E}[Y(1) - Y(0) \mid \|U\| > t], \tag{3.1}$$

which is simply ATE conditioning on large $\|U\|$. However, in the case of extreme effects, the outcome may be unbounded due to the presence of extreme noise. As $t$ increases to infinity, this effect may increase to infinity, making this quantity meaningless. Considering the climate change example, it is possible that dramatic climate change will damage or even destroy human societies, causing the effects of some policies to explode even though the policy can effectively reduce losses and slow down the process. Fortunately, regularly varying distributions have the nice property that as $t$ increases to infinity, $\|U\|/t \mid \|U\| > t$ converges weakly to the Pareto distribution (See Definition 2.3). Inspired by this property, we can normalize the quantity $Y(1) - Y(0) \mid \|U\| > t$ by its growth rate. To characterize the growth of this quantity, we introduce the following polynomial growth assumption.

**Assumption 3.2** (Asymptotic Homogeneous Property). We assume that the covariate $X$ is bounded, i.e. $\|X\| \leqslant R$. Let $f(X, D, U) = \mathbb{E}[Y \mid X, D, U]$. There exists a $L$-Lipschitz continuous function $g(x, d, u)$ and a function $e(t) : \mathbb{R}^+ \to \mathbb{R}^+$ that satisfies $\lim_{t \to \infty} e(t) = 0$ and

$$|\frac{f(x, d, tu)}{t^\alpha} - g(x, d, u)| \leqslant e(t), \ \forall x \in B_R, \ u \in S^{d-1}.$$

This assumption characterizes the growth of the outcome with respect to the extreme noise. In many real-world examples, this assumption is satisfied. For instance, research show that landslide volume often follows a power-law relationship with rainfall intensity Tuganishuri et al. [2024]; the economic loss caused by hurricanes scales polynomially with the maximum wind speed Zhai and Jiang [2014]. In these cases, $f$ grows polynomially with respect to $\|U\|$ and $e(t) = 0$ exactly. We define the Normalized Extreme Treatment Effect (NETE) as

$$\theta^{\text{NETE}} = \lim_{t \to \infty} \mathbb{E}\left[\frac{Y(1) - Y(0)}{t^\alpha} \mid \|U\| > t\right], \tag{3.2}$$

where $\alpha$ is a known index in Assumption 3.2 from prior knowledge. Note that the previous definition (3.1) is a special case of (3.2) when $\alpha = 0$. The intuition for the scaling factor $t^\alpha$ is that under Assumption 3.2, $\mathbb{E}[Y(d)]$ is of the order $O(\|U\|^\alpha)$ and (3.2) is of the order $O(\mathbb{E}[(\|U\|/t)^\alpha \mid \|U\| > t])$, which is finite if $\alpha < \beta$. (3.2) implies that for a large threshold $t$, we have $\mathbb{E}[Y(1) - Y(0)] \approx t^\alpha \theta^{\text{NETE}}$. Therefore, $\theta^{\text{NETE}}$ measures the influence of treatment on the susceptibility of outcome with respect to extreme noise $U$.

We want to remark that NETE naturally sits at the nexus of two well-studied strands of work, tail-conditional expectations in EVT, and average effects or distributional shifts at extreme quantiles, e.g., ATE and QTE. NETE can be understood as a causal analogue of EVT quantity $\mathbb{E}[Z/t \mid Z > t]$, where $Z$ is a regularly varying variable. It generalizes ATE to the setting of extreme events and aligns with the growth rate given by EVT.

## 3.2 Extreme Effect Identification and Estimation

The estimand (3.2) is designed to measure the treatment effect under extreme events, i.e., extremely large $\|U\|$. In practice, there may only be a small fraction of extreme samples in the dataset, which creates difficulties for statistical inference. To efficiently estimate the NETE, we leverage the asymptotic independence property of regularly varying variables (See Definition 2.3) to derive a novel identification formula. In particular, we have the following decomposition.

$$\lim_{t \to \infty} \mathbb{E}\left[\frac{Y(1) - Y(0)}{t^\alpha} \mid \|U\| > t\right] = \lim_{t \to \infty} \mathbb{E}\left[\frac{f(X, 1, U) - f(X, 0, U)}{t^\alpha} \mid \|U\| > t\right] \tag{3.3}$$

$$= \lim_{t \to \infty} \mathbb{E}\left[\frac{f(X, 1, U) - f(X, 0, U)}{\|U\|^\alpha} \cdot \left(\frac{\|U\|}{t}\right)^\alpha \mid \|U\| > t\right]$$

$$= \lim_{t \to \infty} \mathbb{E}\left[g(X, 1, U/\|U\|) - g(X, 0, U/\|U\|) \cdot \left(\frac{\|U\|}{t}\right)^\alpha \mid \|U\| > t\right] \tag{3.4}$$

where we use Assumption 3.2 in the third equality. We can prove that the above quantity equals to

$$\lim_{t \to \infty} \mathbb{E}[g(X, 1, U/\|U\|) - g(X, 0, U/\|U\|) \mid \|U\| > t] \cdot \lim_{t \to \infty} \mathbb{E}[\|U\|^\alpha/t^\alpha \mid \|U\| > t].$$

The first factor measures the average effect of treatment across different directions, while the second factor only depends on the norm of the extreme noise, which can be estimated via standard techniques in extreme value theory. We summarize the identification formula in the following proposition.

**Proposition 3.3** (Identification). Suppose that $U$ is multivariate regularly varying and Assumption 2.1, 2.2, 3.1, 3.2 hold, we have

$$\theta^{\text{NETE}} = \lim_{t \to \infty} \mathbb{E}[g(X, 1, U/\|U\|) - g(X, 0, U/\|U\|) \mid \|U\| > t] \cdot \lim_{t \to \infty} \mathbb{E}[\|U\|^\alpha/t^\alpha \mid \|U\| > t].$$

Proposition 3.3 separates the estimation of NETE into two parts, the expectation of the spectral measure and the index estimation, which facilitates the estimation. While in theory naive identification (3.3) works as well, we found that in practice (3.3) performs poorly (See Section 4 for empirical experiments). One reason is that without properly scaling, the (3.3) suffers from exploding $\|U\|$, causing larger estimation errors.

Inspired by this decomposition, we estimate the two factors separately. To make our framework more flexible, we allow an approximate scaling exponential $\widehat{\alpha}_n$ as input in Algorithm 1. $\widehat{\alpha}_n$ can be obtained from some prior knowledge or via other heuristics. For the first factor, we design two estimators, the Inverse Propensity Weighting (IPW) and the Doubly Robust (DR) estimators. To derive the estimators, we first randomly split the data into equal halves and use the first half for nuisance estimation, i.e., propensity and outcome. Note that the domain of the propensity function is unbounded due to the presence of $U$. Generally, it is difficult to estimate such a function given scarce data in the tail. To bypass this barrier, we make the following independence assumption.

**Assumption 3.4.** (Independence) The extreme variable $U$ is independent of $D, X$, i.e., $U \perp X, D$.

Under Assumption 3.4, only $X$ affects treatment assignment. We use the first half of data to regress $(X, D, U/\|U\|)$ on $Y/\|U\|^{\widehat{\alpha}_n}$ to get (normalized) pseudo-outcome $\widehat{g}$ and regress $X$ on $D$ to get an estimation of the propensity function $\widehat{p}$. Then, we use the second half for estimation. The IPW and DR estimators are defined in (3.5) and (3.6), respectively.

Notice that the second factor is the $\alpha$-moment of the random variable $\|U\|/t \mid \|U\| > t$, which converges weakly to a Pareto distribution as $t$ increases to infinity. Therefore, this quantity equals to the $\alpha$ moment of a standard Pareto $1/(1 - \alpha\gamma)$ and the problem can be reduced to estimating the EVI of an asymptotic Pareto distribution. Here, we use the adaptive Hill estimator in (3.7) from

---

**Algorithm 1** Algorithm for NETE Estimation

---

**Require:** Dataset $\mathcal{D} = \{(X_i, D_i, Y_i, U_i)\}_{i=1}^n$, threshold $t$, exponent estimation $\widehat{\alpha}_n$, estimator
1: Randomly split $\mathcal{D}$ into two equal parts $\mathcal{D}_1$ and $\mathcal{D}_2$
2: Using $\mathcal{D}_1$, estimate:

      a. Propensity function $\widehat{p}(x)$ via regression of $D$ on $X$ and clip the output of $\widehat{p}(x)$ to the interval $[c, 1-c]$.

      b. Pseudo-outcome regression $\widehat{g}(x, d, s)$ by regressing $Y/\|U\|^{\widehat{\alpha}_n}$ on $(X, D, U/\|U\|)$

3: Define index set $\mathcal{I} = \{i : \|U_i\| > t, (X_i, D_i, Y_i, U_i) \in \mathcal{D}_2\}$ and set $S_i = U_i/\|U_i\|$ for $i \in \mathcal{I}$
4: **if** estimator = IPW **then**
5:    Compute

$$\widehat{\eta}_{n,t}^{\text{IPW}} = \frac{1}{|\mathcal{I}|} \sum_{i \in \mathcal{I}} S_i \left( \frac{D_i}{\widehat{p}(X_i)} - \frac{1 - D_i}{1 - \widehat{p}(X_i)} \right). \tag{3.5}$$

6: **else if** estimator = DR **then**
7:    Compute

$$\widehat{\eta}_{n,t}^{\text{DR}} = \frac{1}{|\mathcal{I}|} \sum_{i \in \mathcal{I}} \left[ \widehat{g}(X_i, 1, S_i) - \widehat{g}(X_i, 0, S_i) + \frac{D_i - \widehat{p}(X_i)}{\widehat{p}(X_i)(1 - \widehat{p}(X_i))} \left( Y_i/\|U_i\|^{\widehat{\alpha}_n} - \widehat{g}(X_i, D_i, S_i) \right) \right].$$

$$\tag{3.6}$$

8: **end if**
9: Compute adaptive Hill estimator on $\{\|U_i\| : i \in \mathcal{I}\}$:

$$\widehat{\gamma}_n = \widehat{\gamma}(k) = \frac{1}{k} \sum_{j=1}^{k} \log \frac{\|U_{(j)}\|}{\|U_{(k+1)}\|}, \quad \widehat{\mu}_n = \frac{1}{1 - \widehat{\alpha}_n \widehat{\gamma}_n}, \tag{3.7}$$

   where $\|U_{(1)}\| \geq \cdots \geq \|U_{(k+1)}\|$ and $k$ is chosen by

$$k = \max \left\{ k \in \{l_n, \cdots, n\} \text{ and } \forall i \in \{l_n, \cdots, n\}, |\widehat{\gamma}(i) - \widehat{\gamma}(k)| \leqslant \frac{\widehat{\gamma}(i) r_n(\delta)}{\sqrt{i}} \right\},$$

10: **return** $\widehat{\theta}_{n,t}^{\text{estimator}} = \widehat{\eta}_{n,t}^{\text{estimator}} \cdot \widehat{\mu}_n$.

---

Boucheron and Thomas [2015], which provides a data-driven method for choosing the threshold. Putting the two estimations together, we get our estimator of the NETE $\widehat{\theta}_{n,t} = \widehat{\eta}_{n,t} \cdot \widehat{\mu}_n$, where the superscript $\cdot$ can be DR or IPW. We summarize our estimators in Algorithm 1.

**Remark 3.5.** Assumption 3.4 may not always hold in practice. The identification formula (**??**) still holds regardless of Assumption 3.4. When the assumption is violated, the propensity score based method cannot be applied here. In this case, better estimators may be a simple plug-in estimator or the DR estimator since they do not require accurate propensity estimation.

**Remark 3.6.** The identification formula Proposition 3.3 and our estimation methods can be easily extended for CATE estimation. We will discuss the extension in the appendix.

### 3.3 Non-asymptotic Analysis

Up to now, we have worked under very mild regular variation and asymptotic homogeneity conditions, which suffice to prove the consistency of our two-step estimator in the limit $n, t \to \infty$. However, to obtain non–non-asymptotic, finite-sample deviation bounds for both the spectral-measure term and the tail-index term, we must invoke a more structured tail model. In particular, existing results such as those in Zhang et al. [2023] rely on the fact that, beyond regular variation, the noise vector behaves exactly like a (possibly linearly transformed) Pareto distribution. Although this is admittedly stronger than mere second-order regular variation, it is at present the only framework in which we can directly apply sharp concentration inequalities and Wasserstein-distance bounds for spectral-measure estimation. We therefore make the following Pareto-type assumption.

**Assumption 3.7.** We assume that the distribution of $U$ comes from the following class of models

$$M = \cup_{k=1}^\infty M_k,$$

where $M_k = \{\mathcal{L}(X) : U = AZ,$ for $A \in \mathcal{A}$ and $\mathcal{L}(Z) \in \widetilde{M_k}\}$. The set of possible distributions for the components $Z$ is

$$\widetilde{M_k} = \left\{ \mathcal{L}(Z) : \begin{array}{c} Z \text{ admits a (Lebesgue) density } h(z) \text{in } \mathbb{R}_+^{d_z} \\ \left| \frac{h(z) - \beta^m \prod_{i=1}^m (1+z_i)^{-(\beta+1)}}{\beta^m \prod_{i=1}^m (1+z_i)^{-(\beta+1)}} \right| \leqslant \xi k^{-s}, \forall z \\ h(z) \propto \prod_{i=1}^m (1+z_i)^{-(\beta+1)} \text{if } \|z\|_1 > \zeta k^{\frac{1-2s}{\beta}} \end{array} \right\}$$

and the set of possible matrices $\mathcal{A}$ is

$$\mathcal{A} = \left\{ A \in \mathbb{R}_+^{d_u \times d_z} : l \leq \min_i \|A_{\cdot i}\|_1 \leq \max_i \|A_{\cdot i}\|_1 \leq u \text{ and } JA \geq \sigma \right\}.$$

Throughout, we assume the constants satisfy $m \geq d \geq 2, 0 < l < 1 < u, 0 < s < 1/2, \sigma > 0$, $0 < \xi < 1$, and $\zeta > 0$.

This assumption states that the extreme variable is a linear transformation of an approximate Pareto distribution. The parameter $s$ measures how close $Z$ is to a standard multivariate Pareto distribution. A small $s$ means the distribution is far from Pareto.

With these assumptions, we are ready to state our main theorem, which gives a non-asymptotic rate to our estimand.

**Theorem 3.8.** Suppose that Assumption 2.1, 2.2, 3.1, 3.2, 3.4, 3.7 hold, $\alpha < \beta$, where $\alpha$ and $\beta$ are defined in Assumption 3.2 and Assumption 3.7 respectively. Furthermore, for any fixed $t$, with probability at least $1 - \delta$,

$$|p(X) - \widehat{p}(X)| \leqslant R_p(n, \delta), |\widehat{\alpha}_n - \alpha| \leqslant R_\alpha(n, \delta),$$
$$|\mathbb{E}[Y/\|U\|^\alpha \mid X, D, U/\|U\|, \|U\| > t] - \widehat{g}(X, D, U/\|U\|)| \leqslant R_g(n_t, \delta),$$

where $n_t = \sum_{i=1}^{\lfloor n/2 \rfloor} I(\|U_i\| > t)$ and $R_p, R_g, R_\alpha$ are estimation errors that are monotonically decreasing with respect to sample size. Then, if $n \geqslant \Theta(\log(1/\delta)t^\beta)$, with probability at least $1 - \delta, \delta \in (0, 1/2)$, we have

$$\left| \widehat{\theta}_{n,t}^{\text{DR}} - \theta^{NETE} \right| \leqslant O\left( \sqrt{R_p(n/2, \delta) R_g(nt^{-\beta}, \delta)} + t^{\beta/2} n^{-1/2} + \log(1/\delta) n^{-1/(2+\beta)} \right.$$
$$\left. + t^{-\min\{1,\beta\}} + t^{-\beta s/(1-2s)} + \log(t) R_\alpha(n, \delta) + e(t) \right). \quad (3.8)$$

and

$$\left| \widehat{\theta}_{n,t}^{\text{IPW}} - \theta^{NETE} \right| \leqslant O\left( R_p(n/2, \delta) + t^{\beta/2} n^{-1/2} + \log(1/\delta) n^{-1/(2+\beta)} \right.$$
$$\left. + t^{-\min\{1,\beta\}} + t^{-\beta s/(1-2s)} + \log(t) R_\alpha(n, \delta) + e(t) \right). \quad (3.9)$$

The error bound (3.6) consist of the nuisance error $\sqrt{R_p(n/2, \delta) R_g(nt^{-\beta}, \delta)}$, variance $t^{\beta/2} n^{-1/2}$, EVI estimation error $\log(1/\delta) n^{-1/(2+\beta)}$, $\alpha$ error $R_\alpha(n, \delta)$ and bias terms $t^{-\min\{1,\beta\}} + t^{-\beta s/(1-2s)} + e(t)$. Similar pattern holds for (3.9). Given this general result, we choose the threshold $t$ in a data-driven way to obtain a better rate. The idea is to use the estimated index to balance the bias and variance terms in (3.8) and (3.9). The following corollary gives the convergence rate in two different regimes.

**Corollary 3.9.** Under the assumptions of Theorem 3.8, further suppose that

$$R_p(n, \delta) = \Theta(\log(1/\delta) n^{-1/2}), R_g(n, \delta) = \Theta(\log(1/\delta) n^{-1/2}), R_\alpha(n, \delta) = \Theta(\log(1/\delta) n^{-c_\alpha}),$$

for some $c_\alpha > 0$, the following conclusions hold.

1. If $s \in (0, 1/(2 + \max\{1, \beta\}))$, takes $t_n = \Theta(n^{(1-2s)\widehat{\gamma}_n})$, with probability at least $1 - \delta$, we have
$$|\widehat{\theta}_{n,t}^{\text{DR}} - \theta^{NETE}| = O(e(t_n) + n^{-s} \log(1/\delta) + n^{-c_\alpha} \log(n) \log(1/\delta)).$$

2. If $s \in [1/(2 + \max\{1, \beta\}), 1/2)$, takes $t = \Theta(n^{(\widehat{\gamma}_n/(1+2\min\{1, \widehat{\gamma}_n\}))})$, with probability at least $1 - \delta$, we have
$$|\widehat{\theta}_{n,t}^{\text{DR}} - \theta^{NETE}| = O(e(t_n) + n^{-1/(2+\max\{\beta,1\})} \log(1/\delta) + n^{-c_\alpha} \log(n) \log(1/\delta)).$$

Similar results hold for the IPW estimator. Due to limited space, we leave the result for IPW in the appendix. Many common machine learning algorithms, e.g., Lasso, logistic regression, neural networks, can achieve $O(n^{-1/2})$ rate in the assumption of Corollary 3.9. We want to highlight that if $e(t)$ decays fast enough and become negligible compared to the other term and we know the correct scaling exponential $\alpha$, Corollary 3.9 matches the rate of [Zhang et al., 2023, Theorem 3.1] without prior knowledge on the index $\beta$ in the Assumption 3.7. Besides, if we have additional prior knowledge on $e(t)$ and $c_\alpha$, we can adjust the choice of threshold $t$ to achieve a better rate.

**Remark 3.10.** When the extreme noise is 1-dimensional, the spectral measure is trivially $\delta_{\{1\}}$ and there is no need to estimate the spectral measure. Following a similar argument of Theorem 3.8 and Corollary 3.9, we can obtain a convergence rate of $O(e(t_n) + \log(1/\delta)n^{-1/(2+\beta)} + \log(1/\delta)n^{-c_\alpha})$.

**Remark 3.11.** Assumption 3.7 may seem restricted at first glance. This assumption is used here because the non-asymptotic result for regularly varying extreme distributions is rare in the literature and the goal of this paper is not to develop a new estimator for the spectral measure. To the best of our knowledge, Zhang et al. [2023] is the only paper that gives such a result under Assumption 3.7. In fact, Assumption 3.7 can be easily replaced by the following two assumptions in our proof. (1) The extreme noise $U$ is regularly varying and its norm $\|U\|$ satisfies the von Mises condition in Boucheron and Thomas [2015]. (2) There exists an upper bound for the bias term $\|\mathbb{E}[f(U/\|U\|) \mid \|U\| > t] - \lim_{t\to\infty} \mathbb{E}[f(U/\|U\|) \mid \|U\| > t] \leqslant O(t^{-c_0})$, for some constant $c_0 > 0$ for a fixed Lipschitz function $f$. We leave this generalization to future work.

# 4 Experiments

Having established in Section 3 that under our regularity and overlap assumptions the DR- and IPW-based extreme treatment estimators enjoy a provable non-asymptotic error bound, we next evaluate their finite-sample behavior and compare with our estimators with naive estimators that does not consider the regularly varying structure. In what follows, Section 4.1 presents purely synthetic simulations with known NETE. Section 4.2 then moves to a semi-synthetic setting—using real noise from wavesurge datasets—to assess practical performance under realistic complexities.

## 4.1 Synthetic Dataset

The data generation process we use in this subsection is

$$X \sim \text{Unif}([0,1]^5), D \sim \text{Ber}(p(X)), \text{where } p(x) = 1/(1 + \exp(-x^\mathsf{T} b)),$$

$$Y = \|U\|^\alpha(D + U/\|U\| + \epsilon) + \|U\|^{\alpha/2}, \epsilon \sim \text{Unif}(-1, 1),$$

where $\alpha > 0$ is a constant and $b \sim N(0,1)$, $A \sim \text{Unif}([1,2]^{d_u \times d_z})$. We consider two ways of generating the extreme noise. The first one follows Assumption 3.7.

$$Z = (Z_1, \cdots, Z_{d_z}), Z_i \sim \text{Pareto}(\beta), U = AZ, A \in \mathbb{R}^{d_u \times d_z}.$$

We also consider a Pareto mixture, i.e., $U = (U_1, \cdots, U_{d_u}), U_i \sim 0.5\text{Pareto}(\beta) + 0.5\text{Pareto}(\beta + 1)$. Note that Assumption 3.2 is satisfied with $e(t) = t^{-\alpha/2}$. By Proposition 3.3, we can calculate the ground-truth effect. The graph below shows the Mean Square Error (MSE) with our estimator using different sample sizes. We take different values for $\alpha, \beta$ in the experiments. In this case, by Proposition 3.3, we know that the ground-truth NETE is $1/(1 - \alpha/\beta)$. We use Mean-Square-Error (MSE), $\mathbb{E}[(\widehat{\theta} - \theta^{\text{NETE}})^2]$, to measure the error. As a baseline, we compare our estimator with naive IPW and DR estimators. Naïve-IPW simply applies the standard IPW estimator to the $U_i$ that has norm larger than a threshold $t$, ignoring any tail-index modeling. Similarly, Naïve-DR augments it with the usual doubly-robust correction term but likewise ignores the Pareto structure. We leave the detailed math formulation of the baseline estimators to the appendix. The thresholds rule in Corollary 3.9 is used in the experiments and we use the same threshold selection rules for all estimators. We estimate the scaling exponential $\alpha$ by doing linear regression $\log(|Y|) \sim \log(\|U\|)$ and use the coefficient of $\log(\|U\|)$ as $\widehat{\alpha}_n$. We leave the experiment details to the appendix. Fig. 1 and Fig. 2 show the experiment results. In the following, we use EVT-IPW and EVT-DR to represent $\widehat{\theta}_{n,t}^{\text{IPW}}$ and $\widehat{\theta}_{n,t}^{\text{DR}}$ in Algorithm 1.

Figure 1 and Fig. 2 show that under different configurations of $\alpha, \beta, d_u, d_z$, our estimators generally perform better than the baseline estimators. The reason is that our estimators can make better

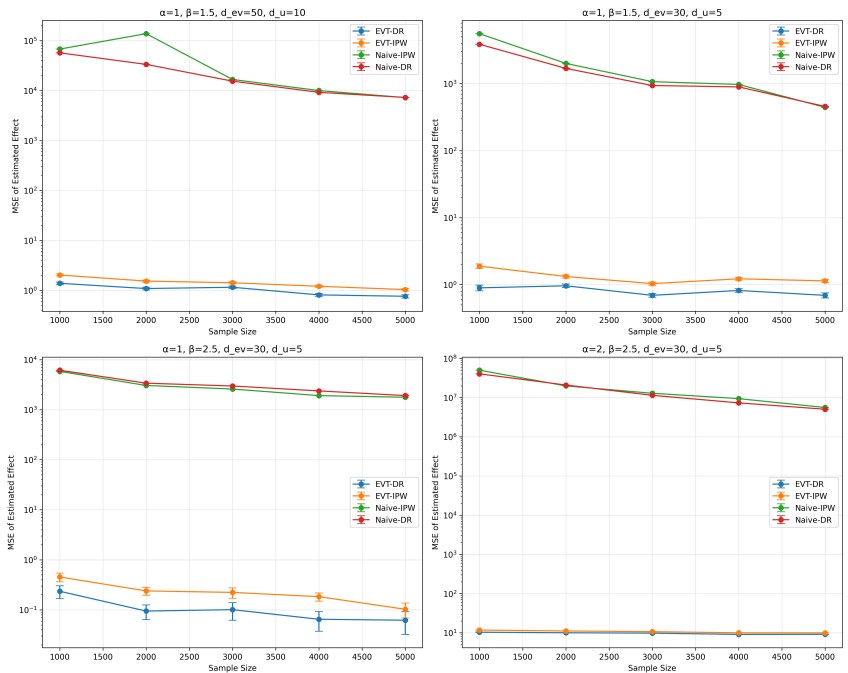

Figure 1: Experiment results of four different configurations when the extreme noise is a linear transformation of Pareto variables. The configures of upper left, upper right, lower left and lower right are $\alpha, \beta, d_z, d_u = (1, 1.5, 50, 10), (1, 1.5, 30, 5), (1, 2.5, 30, 5)$ and $(2, 2.5, 30, 5)$ respectively. The results are averages of 50 repeated experiments. We use EVT-IPW and EVT-DR to represent $\widehat{\theta}_{n,t}^{\mathrm{IPW}}$ and $\widehat{\theta}_{n,t}^{\mathrm{DR}}$ in Algorithm 1.

use of the regularly varying structure. In general, EVT-DR achieves the smallest MSE in most experiments and is robust under different configurations. Note that the Pareto mixture does not satisfy Assumption 3.7. Fig. 2 shows that our method still maintain a good performance even if Assumption 3.7 is violated. We also observe that sometimes the MSE increases with more samples in Fig. 2. An explanation for this is that violation of Assumption 3.7 causes the threshold selection rule in Corollary 3.9 not to be applicable and the variance term dominates the error.

## 4.2 Semi-synthetic Dataset

Now, we use the wavesurge dataset Coles et al. [2001] to create a semi-synthetic dataset for our experiments. The wavesurge dataset has 2894 data points, which contain wave and surge heights at a single location off south-west England. Since wave and surge heights are not in the same scale and may not be positive, we shift the data and normalize each dimension by its $10\%$ quantile. Given the wavesurget dataset, we generate our semi-synthetic dataset in the following way.

$$X \sim \mathrm{Unif}(0, 1), D \sim \mathrm{Ber}(p(X)), \text{where } p(x) = 1/(1 + \exp(-x^{\mathsf{T}} b)),$$
$$Y = (1 - X + D)W^{\alpha_1} S^{\alpha_2} + N(0, 1), \tag{4.1}$$

where $W$ and $S$ are the height of the wave and surge, respectively. In this experiment, we evaluate how well our proposed EVT-based estimators recover the Normalized Extreme Treatment Effect (NETE) when only limited "short-term" data are available. We split the dataset into a training set (1,000 observations) and a test set (1,894 observations). First, we estimate the NETE on the training set using four estimators. Next, we apply the identification formula from Proposition 3.3 together with (4.1) to obtain a high-fidelity estimate of the NETE on the test set. Because the test-set estimate leverages additional data and the correct tail model, we treat it as a surrogate "ground truth" for comparison. The real-world implications of this experiment is that we can use some short-term data (the training set) to predict long-term and unobserved behavior (the test set).

Table 1 shows the results we get using different estimators. The results show that our EVT-IPW and EVT-DR give estimations that are closer to the test-set estimate than the naive estimators. In particular,

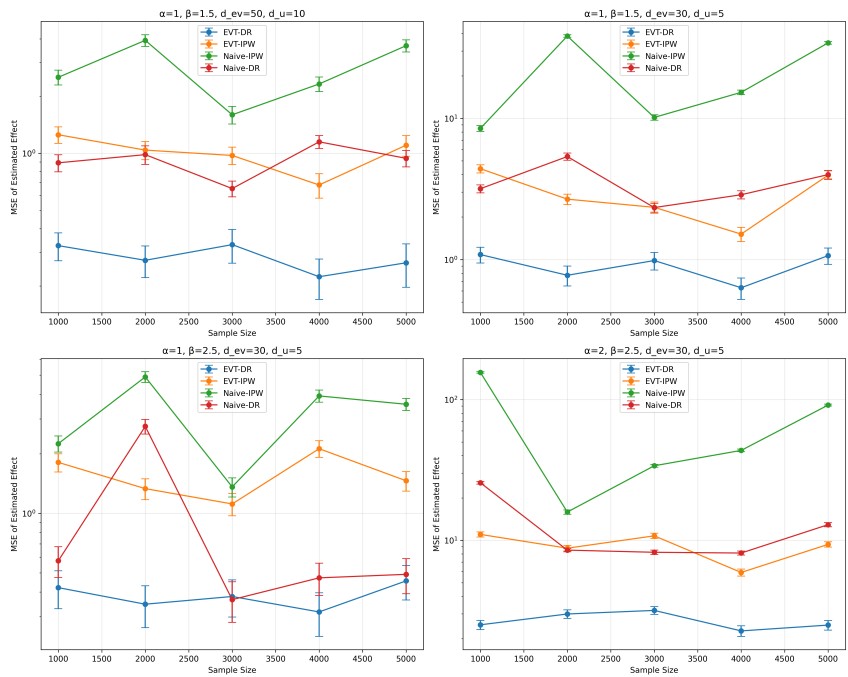

Figure 2: Experiment results of four different configurations when the extreme noise is a Pareto mixture. The configurations of upper left, upper right, lower left and lower right are $\alpha, \beta, d_u = (1, 1.5, 10), (1, 1.5, 5), (1, 2.5, 5)$ and $(2, 2.5, 5)$ respectively. The results are averages of 50 repeated experiments.

the naive estimators consistently overshoot the true NETE by an order of magnitude. In addition, while more extreme tail configurations (e.g. $(1, 3)$) slightly increase variance, the EVT-based methods remain stable, with EVT-DR deviating by at most 0.3 from the test-set estimate. These findings demonstrate that incorporating multivariate extreme value structure via our EVT-IPW and EVT-DR estimators substantially improves finite-sample estimation of treatment effects on rare, tail events, compared both to naive methods.

Table 1: Causal Effect Estimates

| $(\alpha_1, \alpha_2)$ | EVT-DR | EVT-IPW | Naive-DR | Naive-IPW | Test-set Estimate |
|---|---|---|---|---|---|
| $(2, 2)$ | 0.18 | 0.25 | 41.93 | 27.34 | 0.13 |
| $(1, 3)$ | 0.43 | 0.44 | 17.04 | 15.68 | 0.13 |
| $(2.5, 1)$ | 0.13 | 0.18 | 31.64 | 26.06 | 0.20 |
| $(1.5, 1.5)$ | 0.26 | 0.23 | 7.91 | 9.46 | 0.20 |

## 5 Conclusion

In this paper, we tackled the challenge of estimating treatment effects on rare, high-impact events by combining causal inference with extreme value theory. We introduced a new estimand capturing how interventions shift the tail average of outcome distributions and derived a simple identification formula using the spectral–magnitude decomposition of multivariate regular variation. Based on this, we proposed IPW and DR estimators with non-asymptotic error bounds under Pareto-type tails. Simulations and real-data experiments confirmed their advantages over naive estimators when targeting extreme outcomes. This work opens the door to more refined causal analyses in domains such as disaster risk and finance, where tail behavior matters most. A current limitation is our heuristic estimation of the tail index $\alpha$; future work will focus on theoretically grounded estimation of $\alpha$ and adaptive threshold selection.

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

# A    Proofs

## A.1    Identification Formula

In this subsection, we derive the identification formula in Proposition 3.3.

*Proof.* We first prove (3.4). By Assumption 3.2,

$$
\begin{aligned}
\lim_{t \to \infty} \mathbb{E}\left[ \frac{Y(1) - Y(0)}{t^\alpha} \mid \|U\| > t \right] &= \lim_{t \to \infty} \mathbb{E}\left[ \frac{f(X, 1, U) - f(X, 0, U)}{t^\alpha} \mid \|U\| > t \right] \\
&= \lim_{t \to \infty} \mathbb{E}\left[ \frac{f(X, 1, U) - f(X, 0, U)}{\|U\|^\alpha} \cdot \left( \frac{\|U\|}{t} \right)^\alpha \mid \|U\| > t \right] \\
&= \lim_{t \to \infty} \mathbb{E}\left[ (g(X, 1, U/\|U\|) - g(X, 0, U/\|U\|) + 2e(t)) \cdot \left( \frac{\|U\|}{t} \right)^\alpha \mid \|U\| > t \right]
\end{aligned}
$$

We next argue that $\lim_{t \to \infty} \mathbb{E}[(\|U\|/t)^\alpha \mid \|U\| > t] = \alpha/(\beta - \alpha)$. We have

$$
\begin{aligned}
\mathbb{E}[(\|U\|/t)^\alpha \mid \|U\| > t] &= 1 + \int_1^\infty P((\|U\|/t)^\alpha \geqslant r \mid \|U\| > t) \, \mathrm{d}r \\
&= 1 + \frac{\int_1^\infty P(\|U\|/t > r^{1/\alpha}) \, \mathrm{d}r}{P(\|U\| > t)} \\
&= 1 + \frac{\int_1^\infty \alpha P(\|U\| > rt) r^{\alpha-1} \, \mathrm{d}r}{P(\|U\| > t)}
\end{aligned}
$$

Note that $\|U\|$ is also regularly varying. By Potter's theorem [Bingham et al., 1989, Theorem 1.56], for any $\epsilon > 0$ and sufficiently large $t$, we have

$$
\frac{P(\|U\| > rt)}{P(\|U\| > t)} \leqslant 2r^{-\beta+\epsilon}.
$$

Take $\epsilon > 0$ such that $\alpha - \beta + \epsilon - 1 < -1$, we have for sufficiently large $t$,

$$
\frac{\int_1^\infty \alpha P(\|U\| > rt) r^{\alpha-1} \, \mathrm{d}r}{P(\|U\| > t)} \leqslant 2 \int_1^\infty \alpha r^{\alpha-\beta+\epsilon-1} \, \mathrm{d}r < \infty.
$$

Therefore, by the dominance convergence theorem,

$$
\mathbb{E}[(\|U\|/t)^\alpha \mid \|U\| > t] \to 1 + \int_1^\infty \alpha r^{-\beta} \cdot r^{\alpha-1} \, \mathrm{d}r = \beta/(\beta - \alpha),
$$

which implies

$$
\lim_{t \to \infty} e(t) \mathbb{E}[(\|U\|/t)^\alpha \mid \|U\| > t] = 0
$$

and (3.4) holds. We then verify the uniform integrability of function
$$
h(U) = \mathbb{E}_X[(g(X, 1, U/\|U\|) - g(X, 0, U/\|U\|))(\|U\|/t)^\alpha].
$$
Note that by Assumption 3.2, $g$ is a continuous function on a compact set and thus is bounded by some constant $C > 0$. We have
$$
\mathbb{E}[|h(U)| \mid \|U\| > t] \leqslant 2C \mathbb{E}[(\|U\|/t)^\alpha \mid \|U\| > t].
$$

We have proven that the Right Hand Side (RHS) converges to a constant as $t \to \infty$, which implies that $\mathbb{E}[|h(U)| \mid \|U\| > t]$ is uniformly bounded. We conclude that $h(U)$ is uniformly integrable. By the uniform-integrability convergence theorem,

$$
\begin{aligned}
\lim_{t \to \infty} \mathbb{E}\left[ (g(X, 1, U/\|U\|) - g(X, 0, U/\|U\|)) \cdot \left( \frac{\|U\|}{t} \right)^\alpha \mid \|U\| > t \right] &= \mathbb{E}_{(r,\theta) \sim \mathcal{L}}[(g(X, 1, \theta) - g(X, 0, \theta)) r^\alpha] \\
&= \mathbb{E}_{\theta \sim \mathcal{L}}[(g(X, 1, \theta) - g(X, 0, \theta))] \mathbb{E}_{r \sim \mathcal{L}}[r^\alpha].
\end{aligned}
$$

where $\mathcal{L}$ is the limiting distribution of $(\|U\|/t, U/\|U\|) \mid \|U\| > t$ and we use the asymptotic independent property of regularly varying distributions (Definition 2.3). $\qquad\square$

## A.2 Non-asymptotic Analysis

To obtain a convergence rate for the estimator $\widehat{\theta}_n^t$, we first analyze the rate of the two factors $\widehat{\gamma}_n$ and $\widehat{\eta}_{n,t}^{DR}$.

**Lemma A.1.** Under Assumption 3.1, 3.7 with probability at least $1 - \delta$, for sufficiently large $n$, we have

$$|\gamma - \widehat{\gamma}_n| \leqslant O\left(\left(\frac{\log(2/\delta)}{n}\right)^{1/(2+\beta)}\right),$$

where $\gamma = 1/\beta$ is the EVI of $U$.

*Proof.* We adopt the non-asymptotic analysis of the adaptive Hill estimator for EVI in [Boucheron and Thomas, 2015]. In the paper, the author adopts an adaptive estimator, choosing $k$ to be

$$k = \max\left\{k \in \{l_n, \cdots, n\} \text{and} \ \forall i \in \{l_n, \cdots, n\}, |\widehat{\gamma}(i) - \widehat{\gamma}(k)| \leqslant \frac{\widehat{\gamma}(i)r_n(\delta)}{\sqrt{i}}\right\}$$

where $\widehat{\gamma}(i) = \frac{1}{i}\sum_{j=1}^i \log\frac{\|U_{(j)}\|}{\|U_{(i+1)}\|}$ and $r_n(\delta)$ scales like $\sqrt{\log((2/\delta)\log(n))}$. First, we verify the von Mises conditions in Boucheron and Thomas [2015] under Assumption 3.7. Let $F$ be the CDF of $\|U\|$. By Assumption 3.7, we know that

$$g(z) = c\alpha^m \prod_{i=1}^m (1+z_i)^{-\alpha-1}, \|z\|_1 > \zeta k^{(1-2s)/\alpha}.$$

Note that

$$|c - 1| = \left|\frac{g(z) - \alpha^m \prod_{i=1}^m (1+z_i)^{-\alpha-1}}{\alpha^m \prod_{i=1}^m (1+z_i)^{-\alpha-1}}\right| \leqslant \xi k^{-s} \leqslant \xi.$$

Let $\widetilde{Z}_1, \cdots, \widetilde{Z}_m \sim \alpha c^{1/m}(1+z)^{-\alpha-1}, z \geqslant c^{1/(m\alpha)} - 1$. Then, we verify the upper bound for the von Mises function, i.e., $\sup_{s \geqslant t} |\eta(s)| \leqslant O(t^\rho)$ for some $\rho < 0$, where $\eta$ is the von Mises function.

$$\eta(t) = \frac{tU'(t)}{U(t)} - \frac{1}{\beta} \tag{A.1}$$

$$= \frac{1}{tU(t)f(U(t))} - \frac{1}{\beta}, \tag{A.2}$$

where $f(t)$ is the density function of $\sum_{i=1}^m a_i\widetilde{z}_i$. By [Nguyen, 2014, Theorem 2.1], we have that when $\|U\|_1 > \max_i\{a_i\}\zeta k^{(1-2s)/\alpha}$,

$$f(t) = C\beta t^{-\beta-1}(1 + D(1 - 1/\beta)t^{-1} + o(t^{-1})). \tag{A.3}$$

Then,

$$\bar{F}(t) = 1 - F(t) = Ct^{-\beta}(1 + Dt^{-1} + o(t^{-1}))$$

and

$$U(t) = C^{1/\beta}t^{1/\beta}(1 + DC^{-1/\beta}t^{-1/\beta}/\beta + o(t^{-1/\beta})). \tag{A.4}$$

Plug in (A.3) and (A.4) into (A.2), we get

$$\eta(t) = \frac{1}{\beta(1 - DC^{-1/\beta}t^{-1/\beta} + o(t^{1/\beta}))} - \frac{1}{\beta} = O(t^{-1/\beta}).$$

Therefore, the growth rate of the von Mises function is bounded. By [Boucheron and Thomas, 2015], with probability at least $1 - \delta$, we have

$$|\gamma - \widehat{\gamma}_n| \leqslant O\left(\left(\frac{\log(2/\delta)}{n}\right)^{1/(2+\beta)}\right),$$

$\square$

**Lemma A.2.** Undet the assumption of Theorem 3.8, with probability at least $1 - \delta$, we have

$$|\widehat{\eta}_{n,t}^{\mathrm{DR}} - \eta| \leqslant O(\sqrt{R_p(n/2,\delta)R_g(t^{-\beta}n,\delta)} + t^{\beta/2}n^{-1/2} + t^{-\min\{1,\beta\}} + t^{-\beta s/(1-2s)} + e(t) + \log(t)R_\alpha(n,\delta)).$$

*Proof.* Let

$$\eta^t = \mathbb{E}\left[\frac{f(X,1,U) - f(X,0,U)}{\|U\|^{\widehat{\alpha}_n}} \mid \|U\| > t\right],$$

We have the following decomposition

$$|\widehat{\eta}_{n,t}^{\mathrm{DR}} - \eta| \leqslant |\widehat{\eta}_{n,t}^{\mathrm{DR}} - \eta^t| + |\eta^t - \eta|.$$

The first term comes from the standard statistical error of DR estimator, while the second term is the bias term caused by the finite threshold. For the first term, by standard DML theory [Foster and Syrgkanis, 2023], we have

$$|\widehat{\eta}_{n,t}^{\mathrm{DR}} - \eta^t| \leqslant O\left(\sqrt{R_p(n/2,\delta)R_g(n_t,\delta)} + n_t^{-1/2}\right),$$

where $n_t = \sum_{i=1}^{n/2} I(\|U_i\| > t)$ is a random variable. By Bernstein's inequality, with probability at least $1 - \delta$,

$$n_t - n\mathbb{P}(\|U\| > t)/2 \geqslant O\left(\log(1/\delta) + \sqrt{n\mathbb{P}(\|U\| > t)\log(1/\delta)}\right)$$

Therefore, with the same probability, when $n \geqslant \Theta(\log(1/\delta)t^\beta)$.

$$n_t \geqslant \frac{1}{4}n\mathbb{P}(\|U\| > t) = \Theta(nt^{-\beta})$$

and we have

$$|\widehat{\eta}_{n,t}^{\mathrm{DR}} - \eta^t| \leqslant O\left(\sqrt{R_p(n/2,\delta)R_g(nt^{-\beta},\delta)} + t^{\beta/2}n^{-1/2}\right), \tag{A.5}$$

where we use the monotonicity of $R_p, R_g$.

For the second term (the bias term),

$$
\begin{aligned}
|\eta^t - \eta| &= \left|\mathbb{E}\left[\frac{f(X,1,U) - f(X,0,U)}{\|U\|^{\widehat{\alpha}_n}} \mid \|U\| > t\right] - \mathbb{E}[g(X,1,U/\|U\|) - g(X,0,U/\|U\|)]\right| \\
&\leqslant \left|\mathbb{E}\left[\frac{f(X,1,U) - f(X,0,U)}{\|U\|^{\alpha}} - g(X,1,U/\|U\|) - g(X,0,U/\|U\|) \mid \|U\| > t\right]\right| \\
&\quad + |\mathbb{E}[g(X,1,U/\|U\|) - g(X,0,U/\|U\|)] - \mathbb{E}[g(X,1,U/\|U\|) - g(X,0,U/\|U\|) \mid \|U\| > t]| \\
&\quad + \left|\mathbb{E}\left[\frac{f(X,1,U) - f(X,0,U)}{\|U\|^{\alpha}}\left(1 - \|U\|^{\alpha-\widehat{\alpha}_n}\right)\right]\right| \\
&\leqslant 2e(t) + |\mathbb{E}[g(X,1,U/\|U\|) - g(X,0,U/\|U\|)] - \mathbb{E}[g(X,1,U/\|U\|) - g(X,0,U/\|U\|) \mid \|U\| > t]|, \\
&\quad + \left|\mathbb{E}\left[\frac{f(X,1,U) - f(X,0,U)}{\|U\|^{\alpha}}\left(1 - \|U\|^{\alpha-\widehat{\alpha}_n}\right)\right] \mid \|U\| > t\right|
\end{aligned}
$$

where we use Assumption 3.2 in the last equality. By the error rate assumption in Theorem 3.8,

$$
\begin{aligned}
\left|\mathbb{E}\left[\frac{f(X,1,U) - f(X,0,U)}{\|U\|^{\alpha}}\left(1 - \|U\|^{\alpha-\widehat{\alpha}_n}\right)\right] \mid \|U\| > t\right| &\leqslant C\mathbb{E}\left[\left|1 - \|U\|^{\alpha-\widehat{\alpha}_n}\right| \mid \|U\| > t\right] \\
&= O(\log(t)R_\alpha(n,\delta)).
\end{aligned}
$$

Since $g$ is $L$-Lipschitz continuous, the second term is upper bounded by Wasserstein distance $LW_1(\mathcal{L}^t_{U/\|U\|}, \mathcal{L}_{U/\|U\|})$, where $\mathcal{L}^t_{U/\|U\|}$ is the distribution of $U/\|U\|$ conditioning on $\|U\| > t$ and $\mathcal{L}_{U/\|U\|}$ is its limiting distribution as $t \to \infty$. Therefore, we have

$$
\begin{aligned}
|\eta^t - \eta| &\leqslant 2e(t) + LW_1(\mathcal{L}^t_{U/\|U\|}, \mathcal{L}_{U/\|U\|}) + O(\log(t)R_\alpha(n,\delta)) \\
&\leqslant 2e(t) + O(t^{-\min\{1,\beta\}} + t^{-\beta s/(1-2s)} + \log(t)R_\alpha(n,\delta)), \quad\quad (A.6)
\end{aligned}
$$

where we use [Zhang et al., 2023, Proposition 3.1] in the last inequality to upper bound the bias term $W_1(\mathcal{L}^t_{U/\|U\|}, \mathcal{L}_{U/\|U\|})$. Combing (A.5) and (A.6), we get

$$
|\widehat{\eta}^t_n - \eta| \leqslant O(\sqrt{R_p(n/2,\delta)R_g(t^{-\beta}n,\delta)} + t^{\beta/2}n^{-1/2} + t^{-\min\{1,\beta\}} + t^{-\beta s/(1-2s)} + e(t) + \log(t)R_\alpha(n,\delta)).
$$

$\square$

**Lemma A.3.** Under the assumption of Theorem 3.8, with probability at least $1 - \delta$, we have

$$
|\widehat{\eta}^{\mathrm{IPW}}_{n,t} - \eta| \leqslant O(R_p(n/2,\delta) + t^{\beta/2}n^{-1/2} + t^{-\min\{1,\beta\}} + t^{-\beta s/(1-2s)}) + e(t).
$$

*Proof.* Similar to Lemma A.2, we have the following decomposition.

$$
|\widehat{\eta}^{\mathrm{IPW}}_{n,t} - \eta| \leqslant |\widehat{\eta}^{\mathrm{IPW}}_{n,t} - \eta^t| + |\eta^t - \eta|.
$$

Term $|\eta^t - \eta|$ can be bounded in the same way as in the proof of Lemma A.2. By [Su et al., 2023, Theorem 1], we have

$$
|\widehat{\eta}^{\mathrm{IPW}}_{n,t} - \eta^t| \leqslant O(R_p(n/2,\delta) + n_t^{-1/2}) = O(R_p(n/2,\delta) + t^{\beta/2}n^{-1/2} + \log(t)R_\alpha(n,\delta)).
$$

The rest of the proof is similar. $\square$

Now we are ready to prove Theorem 3.8.

*Proof of Theorem 3.8.* Note that by the asymptotic independence property of regularly varying distribution,

$$
\begin{aligned}
|\widehat{\theta}^{\mathrm{DR}}_{n,t} - \theta^{\mathrm{NETE}}| &= |\widehat{\eta}^{\mathrm{DR}}_{n,t} \cdot \widehat{\mu}_n - \eta \cdot \mu| \\
&\leqslant |\mu| \cdot |\widehat{\eta}^{\mathrm{DR}}_{n,t} - \eta| + |\widehat{\eta}^{\mathrm{DR}}_{n,t}| \cdot |\widehat{\mu}_n - \mu|. \quad\quad (A.7)
\end{aligned}
$$

By Lemma A.1 and A.2, with high probability, $\widehat{\gamma}_n$ and $|\widehat{\eta}^{\mathrm{DR}}_{n,t}|$ is bounded. Note that by Lemma A.1,

$$
|\widehat{\mu}_n - \mu| = \left| \frac{1}{1 - \widehat{\alpha}_n \widehat{\gamma}_n} - \frac{1}{1 - \alpha\gamma} \right| = O(|\widehat{\alpha}_n - \alpha| + |\widehat{\gamma}_n - \gamma|) = O(\log(1/\delta)n^{-1/(2+\beta)} + R_\alpha(n,\delta))
$$

Therefore, by Lemma A.2 and (A.7),

$$
\begin{aligned}
|\widehat{\theta}^{\mathrm{DR}}_{n,t} - \theta^{\mathrm{NETE}}| \leqslant O(&\sqrt{R_p(n/2,\delta)R_g(nt^{-\beta},\delta)} + t^{\beta/2}n^{-1/2} + \log(1/\delta)n^{-1/(2+\beta)} \\
&+ t^{-\min\{1,\beta\}} + t^{-\beta s/(1-2s)} + \log(t)R_\alpha(n,\delta) + e(t)).
\end{aligned}
$$

The bound for $\widehat{\theta}^{\mathrm{IPW}}_{n,t}$ can be proven similarly. $\square$

**Corollary A.4** (Convergence rate for IPW). Under the assumptions of Theorem 3.8, further suppose that

$$
R_p(n,\delta) = \Theta(\log(1/\delta)n^{-1/2}), R_g(n,\delta) = \Theta(\log(1/\delta)n^{-1/2}), R_\alpha(n,\delta) = \Theta(\log(1/\delta)n^{-c_\alpha}),
$$

for some $c_\alpha > 0$, the following conclusions hold.

1. If $s \in (0, 1/(2 + \max\{1,\beta\}))$, takes $t_n = \Theta(n^{(1-2s)\widehat{\gamma}_n})$, with probability at least $1 - \delta$, we have

$$
|\widehat{\theta}^{\mathrm{IPW}}_{n,t} - \theta^{NETE}| = O(e(t_n) + n^{-s}\log(1/\delta) + n^{-c_\alpha}\log(n)\log(1/\delta)).
$$

2. If $s \in [1/(2 + \max\{1, \beta\}), 1/2)$, takes $t = \Theta(n^{(\widehat{\gamma}_n/(1+2\min\{1,\widehat{\gamma}_n\}))})$, with probability at least $1 - \delta$, we have

$$|\widehat{\theta}_{n,t}^{\text{IPW}} - \theta^{NETE}| = O(e(t_n) + n^{-1/(2+\max\{\beta,1\})}\log(1/\delta) + n^{-c_\alpha}\log(n)\log(1/\delta)).$$

*Proof of Corollary 3.9.* By Theorem 3.8 and the error rate assumption in Corollary 3.9, we have

$$\left|\widehat{\theta}_{n,t}^{\text{DR}} - \theta^{NETE}\right| \leqslant O(\log(1/\delta)t^{\beta/4}n^{-1/2} + t^{\beta/2}n^{-1/2} + \log(1/\delta)n^{-1/(2+\beta)}$$
$$+ t^{-\min\{1,\beta\}} + t^{-\beta s/(1-2s)} + \log(t)\log(1/\delta)n^{-c_\alpha} + e(t))$$
$$\leqslant O(\log(1/\delta)t^{\beta/2}n^{-1/2} + \log(1/\delta)n^{-1/(2+\beta)}$$
$$+ t^{-\min\{1,\beta\}} + t^{-\beta s/(1-2s)} + \log(t)\log(1/\delta)n^{-c_\alpha} + e(t)).$$

If $s \in (0, 1/(2 + \max\{1, \beta\}))$, we have

$$\left|\widehat{\theta}_{n,t}^{\text{DR}} - \theta^{NETE}\right| \leqslant O(\log(1/\delta)t^{\beta/2}n^{-1/2} + \log(1/\delta)n^{-1/(2+\beta)}$$
$$+ t^{-\beta s/(1-2s)} + \log(t)\log(1/\delta)n^{-c_\alpha} + e(t)).$$

Takes $t_n = \Theta(n^{(1-2s)\widehat{\gamma}_n})$, we get

$$\left|\widehat{\theta}_{n,t_n}^{\text{DR}} - \theta^{NETE}\right| \leqslant O(\log(1/\delta)n^{(1-2s)\widehat{\gamma}_n\beta/2-1/2} + \log(1/\delta)n^{-1/(2+\beta)}$$
$$+ n^{-\widehat{\gamma}_n\beta s} + \log(t)\log(1/\delta)n^{-c_\alpha} + e(t_n)). \tag{A.8}$$

By Lemma A.1, we have

$$|\widehat{\gamma}_n - \gamma| = |\widehat{\gamma}_n - 1/\beta| \leqslant O(\log(1/\delta)n^{-1/(2+\beta)}).$$

Therefore, $n^{\widehat{\gamma}_n\beta} = 1 + O(n^{-1/(2+\beta)})$. Plug this bound into (A.8) and we can get the results. Similarly, if $s \in [1/(2 + \max\{1, \beta\}), 1/2)$, we have

$$\left|\widehat{\theta}_{n,t}^{\text{DR}} - \theta^{NETE}\right| \leqslant O(\log(1/\delta)t^{\beta/2}n^{-1/2} + \log(1/\delta)n^{-1/(2+\beta)}$$
$$+ t^{-\min\{1,\beta\}} + \log(t)\log(1/\delta)n^{-c_\alpha} + e(t)).$$

Take $t_n = \Theta(\widehat{\gamma}_n/(1 + 2\min\{1, \widehat{\gamma}_n\}))$, we get the results. $\square$

# B  Experiment

In this section, we introduce some details of our experiments and provide additional experiments regarding the sensitivity of the algorithm with respect to the scaling parameter $\alpha$.

## B.1  Implementation Details

The two baseline estimators we consider are naive-IPW:

$$\widehat{\theta}_{n,t}^{\text{Naive-IPW}} = \frac{1}{t^\alpha n_t} \sum_{i>n/2:\|U_i\|\geqslant t} Y_i\left(\frac{D_i}{\widehat{p}(X_i)} - \frac{1-D_i}{1-\widehat{p}(X_i)}\right).$$

and naive-DR:

$$\widehat{\eta}_{n,t}^{\text{Naive-DR}} = \frac{1}{t^\alpha n_t} \sum_{i>n/2:\|U_i\|\geqslant t} \left[\widehat{g}(X_i, 1, U_i) - \widehat{g}(X_i, 0, U_i) + \frac{D_i - \widehat{p}(X_i)}{\widehat{p}(X_i)(1-\widehat{p}(X_i))}\left(Y_i - \widehat{g}(X_i, D_i, U_i)\right)\right],$$

where $n_t = \sum_{i=\lfloor n/2\rfloor+1}^n I(\|U_i\| \geqslant t)$ and $\widehat{p}(X)$ and $\widehat{g}(\cdot)$ are the estimated propensity function and the outcome function respectively. The nuisance estimation of $\widehat{g}$ is obtained by running a regression $Y \sim (X, D, U)$. We clip the propensity to $[10^{-4}, 1 - 10^{-4}]$ to ensure the overlap assumption (Assumption 2.2). $\epsilon \sim \text{Unif}([-1, 1])$ in the data generation in synthetic experiments. We use sample splitting in our experiment, using the first half for nuisance estimation. In the experiment, we use the same threshold $t$ for all estimators, which is given by Corollary 3.9. To choose the threshold, we

first use the adaptive Hill estimator Boucheron and Thomas [2015] to get an estimation of EVI $\widehat{\gamma}_n$ and then set the threshold to be $t = 0.25n^{(\widehat{\gamma}_n/(1+2\min\{1,\widehat{\gamma}_n\}))}$ as in Theorem 3.8. The approximate exponential $\widehat{\alpha}_n$ is coefficient of $\log(\|U\|)$ in linear regression $\log(|Y|) \sim \log(\|U\|)$. For the adaptive Hill estimator Boucheron and Thomas [2015], we follow authors' choice for hyperparameters and choose $l_n = 30, r(\delta) = \sqrt{\log\log(n)}$ and

$$k = \min\left\{k \in \{l_n, \cdots, n\} \text{ and } \exists\, i \in \{l_n, \cdots, n\}, |\widehat{\gamma}(i) - \widehat{\gamma}(k)| > \frac{\widehat{\gamma}(i)r_n(\delta)}{\sqrt{i}}\right\} - 1,$$

where $\widehat{\gamma}(i) = \frac{1}{i}\sum_{j=1}^{i}\log\frac{\|U_{(j)}\|}{\|U_{(i+1)}\|}$.

We run logistic regression to estimate the propensity function and use random forest to model the outcome.

For the semi-synthetic experiment, we apply the same hyperparameter as above to estimate NETE. We shift the data to make it positive and normalize each dimension by its 10 % quantile. The Fig. 3 shows the rough distribution of the wavesurge data after these transformations.

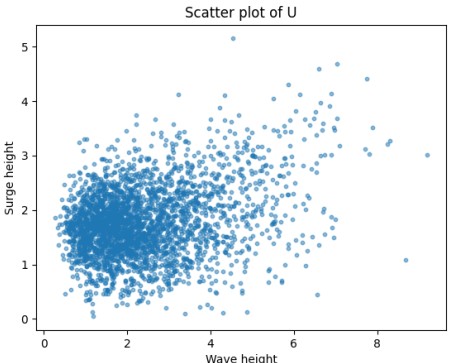

Figure 3: The scatter plot of wavesurge data.

We now describe how we calculate test-set estimation in our experiments. By the data generation process,

$$\theta^{\text{NETE}} = \lim_{t\to\infty} \mathbb{E}[\frac{W^{\alpha_1}S^{\alpha_2}}{t^{\alpha_1+\alpha_2}} \mid \|U\| > t]$$
$$= \lim_{t\to\infty} \mathbb{E}[\frac{W^{\alpha_1}S^{\alpha_2}}{\|U\|^{\alpha_1+\alpha_2}} \mid \|U\| > t] \cdot \frac{1}{1-(\alpha_1+\alpha_2)\gamma},$$

where we use Proposition 3.3 in the second equality. We know the ground-truth $\alpha_1, \alpha_2$ and we can estimate the EVI $\gamma$ using the test set. Suppose the estimated EVI is $\widehat{\gamma}$, we set the threshold to $t_n = 0.25n^{(\widehat{\gamma}/(1+2\min\{1,\widehat{\gamma}\}))}$ and get estimation

$$\widehat{\theta}_{\text{test}}^{\text{NETE}} = \mathbb{E}_n[\frac{W^{\alpha_1}S^{\alpha_2}}{\|U\|^{\alpha_1+\alpha_2}} \mid \|U\| > t_n] \cdot \frac{1}{1-(\alpha_1+\alpha_2)\widehat{\gamma}}.$$

## B.2  Additional Experiments

We conduct additional experiments under the same setup as in Section 4. Fig. 4 and Fig. 5 show the comparison of EVT-IPW and EVT-DR with and without ground-truth $\alpha$. We can see that in most cases, their performance is similar, indicating that our heuristic method for estimating $\alpha$ is effective. Indeed, the heuristic approach even gives slightly better MSE.

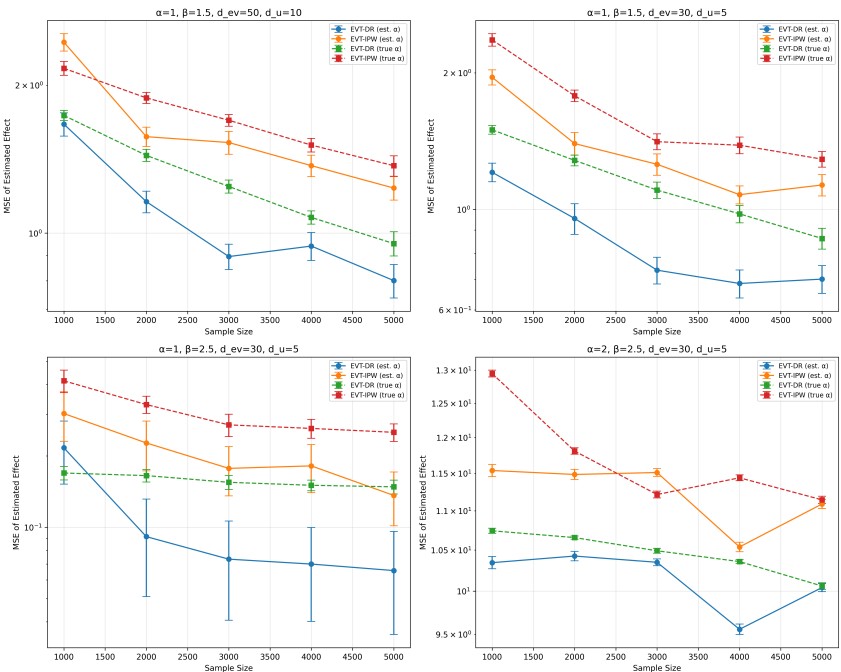

Figure 4: Experiment results of four different configurations when the extreme noise is a linear transformation of Pareto variables. The configures of upper left, upper right, lower left and lower right are $\alpha, \beta, d_z, d_u = (1, 1.5, 50, 10), (1, 1.5, 30, 5), (1, 2.5, 30, 5)$ and $(2, 2.5, 30, 5)$ respectively. The results are averages of 50 repeated experiments.

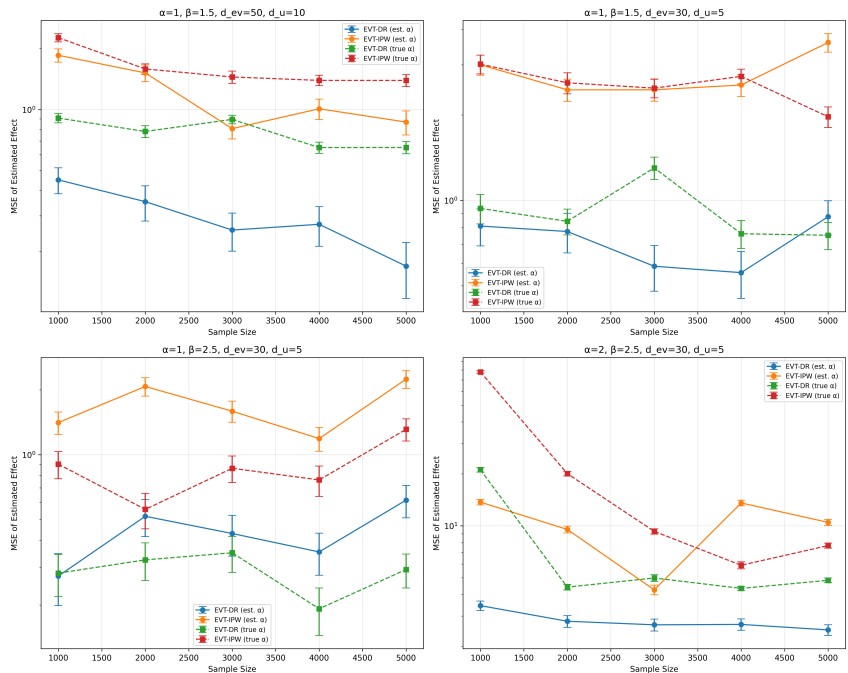

Figure 5: Experiment results of four different configurations when the extreme noise is a Pareto mixture. The configures of upper left, upper right, lower left and lower right are $\alpha, \beta, d_z, d_u = (1, 1.5, 50, 10), (1, 1.5, 30, 5), (1, 2.5, 30, 5)$ and $(2, 2.5, 30, 5)$ respectively. The results are averages of 50 repeated experiments.

## C  Extension: Extreme Normalized Heterogeneous Effect Estimation

**Proposition C.1** (Identification)**.** Suppose that $U$ is multivariate regularly varying and Assumption 2.1, 2.2, 3.1, 3.2 hold, we have

$$\theta^{\mathrm{CNETE}}(x) = \lim_{t \to \infty} \mathbb{E}[g(x, 1, U/\|U\|) - g(x, 0, U/\|U\|) \mid \|U\| > t] \cdot \lim_{t \to \infty} \mathbb{E}[\|U\|^{\alpha}/t^{\alpha} \mid \|U\| > t].$$

*Proof.* The proof is similar to the proof of Proposition 3.3. By Assumption 3.2,

$$
\begin{aligned}
\lim_{t \to \infty} \mathbb{E}\left[\frac{Y(1) - Y(0)}{t^{\alpha}} \mid X = x, \|U\| > t\right] &= \lim_{t \to \infty} \mathbb{E}\left[\frac{f(x, 1, U) - f(x, 0, U)}{t^{\alpha}} \mid \|U\| > t\right] \\
&= \lim_{t \to \infty} \mathbb{E}\left[\frac{f(x, 1, U) - f(x, 0, U)}{\|U\|^{\alpha}} \cdot \left(\frac{\|U\|}{t}\right)^{\alpha} \mid \|U\| > t\right] \\
&= \lim_{t \to \infty} \mathbb{E}\left[(g(x, 1, U/\|U\|) - g(x, 0, U/\|U\|) + 2e(t)) \cdot \left(\frac{\|U\|}{t}\right)^{\alpha} \mid \|U\| > t\right].
\end{aligned}
$$

By the proof of Proposition 3.3, we know $\lim_{t \to \infty} \mathbb{E}\left[(\|U\|/t)^{\alpha} \mid \|U\| > t\right] = \alpha/(\beta - \alpha)$. Thus,

$$\lim_{t \to \infty} \mathbb{E}\left[\frac{Y(1) - Y(0)}{t^{\alpha}} \mid X = x, \|U\| > t\right] = \lim_{t \to \infty} \mathbb{E}\left[(g(x, 1, U/\|U\|) - g(x, 0, U/\|U\|)) \cdot \left(\frac{\|U\|}{t}\right)^{\alpha} \mid \|U\| > t\right].$$

Following the same argument, one can prove the RHS is uniformly bounded. By the dominance convergence theorem,

$$
\begin{aligned}
\lim_{t \to \infty} \mathbb{E}\left[\frac{Y(1) - Y(0)}{t^{\alpha}} \mid X = x, \|U\| > t\right] &= \mathbb{E}_{(r,\theta) \sim \mathcal{L}}\left[(g(x, 1, \theta) - g(x, 0, \theta))r^{\alpha}\right] \\
&= \mathbb{E}_{\theta \sim \mathcal{L}}[g(x, 1, \theta) - g(x, 0, \theta)]\mathbb{E}_{r \sim \mathcal{L}}\left[r^{\alpha}\right].
\end{aligned}
$$

$\square$

Similar to NETE estimation, we can separate the estimation of CNETE into two parts. The scaling factor $\mathbb{E}[\|U\|^{\alpha}/t^{\alpha} \mid \|U\| > t]$ is the same as the factor in (A.7). For the spectral part, one can use meta learners, e.g, X-learner, S-learner, T-learner, DR-learner, for estimation.

