# OpenReview forum: "Estimation of Treatment Effects in Extreme and Unobserved Data"
_NeurIPS.cc/2025/Conference — NeurIPS 2025 poster_

### Official Review · Reviewer_iYgk · 2025-06-02

**Clarity:** 3
**Significance:** 3
**Originality:** 3
**Rating:** 4
**Confidence:** 4

**Summary:**

This paper introduced the Normalized Extreme Treatment Effect (NETE), and then studied its identification and estimation problems.

**Questions:**

Assumption 3.1 appears somewhat restrictive. It is reasonable to expect that the vector of maximum wind speed, rainfall, and storage would influence the level of infrastructure investment. The variable U could serve as a confounder. Can U be included in the covariate vector X to control for its confounding effects?

**Ethical Concerns:**

["NO or VERY MINOR ethics concerns only"]

**Final Justification:**

I will keep my positive score.

**Limitations:**

yes

**Quality:**

3

**Strengths And Weaknesses:**

**Strength**

This paper is theoretically well-founded. The NETE represents a particularly interesting quantity in the context of causal inference. I have no concerns regarding the theoretical development presented in the paper.

**Waekness**

1. My main concern lies in the explanation of the NETE. While these quantities may represent interesting measures in causal inference, the current exposition is limited. To enhance clarity and interpretability, I suggest providing a more detailed explanation, ideally supported by visual illustrations. For instance, consider the joint distribution of (Y_1, Y_0, U) or (Y_1 - Y_0, U): under what conditions does the NETE attain large or small values? Plots or density functions illustrating these relationships would be highly informative.

2. Furthermore, the explanation of the results for the semi-synthetic dataset is limited. In most parts of the semi-synthetic application, the focus is primarily on the statistical properties of the estimators. However, more detailed interpretation of the results is needed to enhance the reader’s understanding of their practical implications.


3. From a statistical perspective, extreme value theory is well-established and well-motivated. However, researchers in the machine learning or causal inference community may be less familiar with this framework. Therefore, a more accessible explanation and stronger motivation would help broaden the impact and understanding of the work. In particular, the meaning of \alpha may be important and warrants further clarification.

4. Assumption 3.1 appears somewhat restrictive. (See question)

---

> ### Author Rebuttal · Authors · 2025-07-31
>
> Thanks for the valuable feedback! Please see below for our detailed responses. We hope our responses can resolve your concern.
>
> 1. Assumption 3.1 appears somewhat restrictive. Can U be included in the covariate vector X to control for its confounding effects?
>
> Thanks for the insightful question! The short answer is that we can include U in the covariate vector, but for estimation, we need to adapt our estimators. Following the same argument as Proposition 3.3, one can derive a similar identification formula. However, when we include U in the covariates, the domain of propensity is not compact. It is difficult to estimate the propensity function defined in the whole real space. If we don’t have access to the propensity function, which sometimes can be obtained from policy documents, we suggest using the DR estimator or the regression adjustment estimator for estimation in this setting.
>
> 2. My main concern lies in the explanation of the NETE. Provide a more detailed explanation, ideally supported by visual illustrations.
>
>
> Thanks for the valuable suggestion. We now include a more detailed, intuitive discussion of the NETE in a simple setting:
>
>  **Data‐Generating Process**: Consider no covariates and let \(U\) be a univariate Pareto with extreme value index $1/\beta $. We generate potential outcomes as
>    $$Y(d) = \bigl(d\theta_{1} + (1-d)\theta_{0}\bigr)\,U^{2}+ 0.1\,\sqrt{U}+ \varepsilon,\quad\varepsilon \sim N(0,\,0.1),$$
>    where \(0 < \theta_{0} < \theta_{1}\).  In this case, $\alpha = 2$ so that
>    $$
>    E\bigl[Y(d)/t^{2}\mid U > t\bigr]\sim c_{d} t^{2}
>    $$
>    for constants $c_{d}$.  By definition, the NETE is
>    $$
>    \mathrm{NETE} = c_{1} - c_{0}= (\theta_{1} - \theta_{0}) \cdot \frac{\beta}{\beta - 1}.
>    $$
>    Intuitively, NETE measures how much faster the treated outcome grows in the extreme tail compared to the control.
>
>  **Interpretation** :
>    - A **large positive NETE** occurs when:
>      1. $\theta_{1} - \theta_{0}$ is large (treatment increases the leading coefficient), or
>      2. $\beta$ is close to 1 (Pareto tail is very heavy).
>    - These two factors correspond to:
>      1. **Faster tail‐growth under treatment**, and
>      2. **Longer tail of $U$**.
>
>  **Planned Illustration** :We will add a figure plotting 10,000 simulated points of $\bigl(Y(1),U\bigr)$ and $\bigl(Y(0),U\bigr)$ with
>    $\beta = \tfrac{10}{11}$, $\theta_{0} = 0.5$, and $\theta_{1} = 1$.  The scatter will fan out around the curves $Y(d)=\theta_{d}\,U^{2}$.  A steeper “fan” visually conveys a larger NETE, illustrating how the joint behavior of $(Y(d),U)$ encodes tail sensitivity.
>
> *We were unable to upload the figure here, but will include it in the revised manuscript.*
>
>
> 3. Furthermore, the explanation of the results for the semi-synthetic dataset is limited. In most parts of the semi-synthetic application, the focus is primarily on the statistical properties of the estimators. However, more detailed interpretation of the results is needed to enhance the reader’s understanding of their practical implications.
>
> Thanks for the suggestion. We will add result analysis in later versions. We can see from equation (4.1) that the treatment $D$ will only change the coefficient of $W^{\alpha_1}S^{\alpha_2}$, which is proportional to the NETE. As mentioned in the previous responses, a positive NETE means the treatment will make the tailed expected outcome grow faster with respect to the threshold t.
>
> 4. From a statistical perspective, extreme value theory is well-established and well-motivated. However, researchers in the machine learning or causal inference community may be less familiar with this framework. Therefore, a more accessible explanation and stronger motivation would help broaden the impact and understanding of the work. In particular, the meaning of \alpha may be important and warrants further clarification.
>
> Thanks for the advice. We will add an extra section in the appendix to briefly introduce the extreme value theory. Besides, we will also add a paragraph in the introduction part to motivate EVT.

---

> > ### Comment · Reviewer_iYgk · 2025-08-03
> >
> > Thank you for your response. I will keep my score.

---

### Official Review · Reviewer_jd2q · 2025-06-20

**Clarity:** 3
**Significance:** 3
**Originality:** 4
**Rating:** 5
**Confidence:** 3

**Summary:**

The paper tackles the challenging problem of estimating the impact of an intervention on rare but extreme outcomes, like extreme hurricanes or other climate events. The authors do so by combining tools from both causal inference and extreme value theory (EVT). They make several contributions:

* Define key estimands when there is extreme tail risk, as well as corresponding identification and assumptions

* Propose an algorithm for estimating these effects, combining both causal ML methods and EVT tools

* A non-asymptotic analysis of the proposed algorithm

* Several experiments and a semi-synthetic application

**Questions:**

* I had a hard time following the discussion in Sec 2 and 3.1 that defines the key quantities of interest, and especially the relationship between U and the other variables. The most intuitive explanation is the sentence on line 124 “for example …”. But it would be useful to give more clarity on the relationships between U and Y especially. e.g., do you think of Y as having a structural component and U noise? The synthetic dataset in Sec 4.1 is a useful step.

* For the semi-synthetic dataset, does $Y$ correspond to anything concrete? Is there any notion of uncertainty that can help calibrate whether these point estimates are “large” or “small”?

**Ethical Concerns:**

["NO or VERY MINOR ethics concerns only"]

**Final Justification:**

Keeping my positive score.

**Limitations:**

yes

**Quality:**

3

**Strengths And Weaknesses:**

Strengths:
* The paper rightly highlights the need for this type of approach. The paper then does an excellent job of bridging the otherwise distinct fields of causal inference and EVT, and gives a very thoughtful discussion of the benefits and drawbacks of each approach.

* The proposed algorithm and corresponding technical analysis seem to be state-of-the-art, and seem to perform well in experiments.

* The paper is up front about remaining challenges and suggests many useful directions for future work.


Weaknesses:
* The appendix appears to be missing, which makes review more challenging.

* The relationship between $Y$ and $U$ needs further clarification (see below).

* Recognizing that this is challenging, there is no discussion of uncertainty quantification, which would be particularly important in a setting like this.

---

> ### Author Rebuttal · Authors · 2025-07-31
>
> Thanks for your valuable review of our paper! We are sure your feedback will make our paper better! Please see the following paragraphs for our responses.
>
> 1. The appendix appears to be missing, which makes the review more challenging.
>
> Thanks for asking. The appendix is in the supplementary material.
>
> 2. It would be useful to give more clarity on the relationships between U and Y, especially. e.g., do you think of Y as having a structural component and U noise? The synthetic dataset in Sec 4.1 is a useful step.
>
> Thanks for asking! We agree that the name “extreme noise” can be confusing. We will make the discussion clearer in later versions. As in the example of line 124, we think U is also a structural component of the model, which represents extreme events. A more appropriate name for U may be “extreme factors”.
>
> 3. For the semi-synthetic dataset, does it correspond to anything concrete? Is there any notion of uncertainty that can help calibrate whether these point estimates are “large” or “small”?
>
> Thanks for the question. In the semi-synthetic dataset, we use real-world data as the extreme noise U and generate the rest of the variables by equations (4.1). (4.1) does not correspond to a real-world example. We use this example to conceptually verify our theoretical results. For uncertainty quantification, please see our response for the next question.
>
> 4. Recognizing that this is challenging, there is no discussion of uncertainty quantification, which would be particularly important in a setting like this.
>
> We thank the reviewer for highlighting the absence of uncertainty quantification. We agree that assessing the variability of our extreme‐treatment estimates is both important and nontrivial. A natural approach to estimating the sampling variability of our estimators is the bootstrap [1]. However, straightforward resampling of extreme observations can fail to reproduce tail dependence and may severely underestimate variance [2]. Recent work on adapting bootstrap methods to heavy‐tailed and extreme‐value settings [3] offers promising strategies—for instance, resampling only the exceedances above a high threshold or applying a parametric tail bootstrap based on the generalized Pareto approximation. While these approaches suggest viable adaptations, a full theoretical justification in our context would require substantial new analysis. We therefore view rigorous confidence‐interval construction for extreme‐treatment effects as a direction for future work and refrain from making formal claims about interval validity in the current manuscript.
>
> [1] Efron, B. (1979). Bootstrap methods: Another look at the jackknife. The Annals of Statistics, 7(1), 1–26.
>
> [2] Cornea‐Madeira & Davidson (2015). “A Parametric Bootstrap for Heavy-Tailed Distributions.” Econometric Theory 31(3): 449–470.
>
> [3] de Haan, Laurens, and Chen Zhou. "Bootstrapping extreme value estimators." Journal of the American Statistical Association 119.545 (2024): 382-393.

---

### Official Review · Reviewer_U7z2 · 2025-06-27

**Clarity:** 4
**Significance:** 3
**Originality:** 3
**Rating:** 4
**Confidence:** 4

**Summary:**

The authors propose a principled approach to estimate causal effects conditional on threshold exceedances in a heavy-tailed data-generating process. Their method is based on a model in which data is generated as a function of a Pareto-type intensity random variable. The causal effect is quantified as the rescaled difference in means between treatment groups. The authors introduce multiple estimators and analyze their non-asymptotic theoretical performance. These results are illustrated through extensive experiments comparing the performance of the proposed estimators for the so-called NETE.

**Questions:**

- Would it be possible to provide deeper insight into what the NETE represents and what type of causal relationship it captures?
- Similarly, could a synthetic experiment be constructed where the NETE identifies a causal relationship that would not be detected by a classical causal inference method (again, assuming the mean exists)?
- Where is the proof for Proposition 3.3? Not sure see why this holds.

More generally:
- Line 13+: The term “extremities” is somewhat awkward, as it can imply an upper bound. Consider using “extremes” or “extreme events” instead.
- Line 84: The term “consistency” is unclear in this context. Please clarify whether it refers to consistency of an estimator, or something else.
- Lines 94 and 104: Is this the same constant c? If so, it should be explicitly stated; if not, the notation should be differentiated.
- Lines 91 and 115: The mean is used before it is formally defined. Consider introducing the necessary definitions earlier for clarity.
- Line 91: The orthogonality symbol is used without explanation. Please define this notation when it first appears.
- Line 94: The term “estimated propensity” is vague—does it refer to an estimated propensity score? If so, please make this explicit.

**Ethical Concerns:**

["NO or VERY MINOR ethics concerns only"]

**Final Justification:**

I maintain my score as explained in the below comment and sincerly thank the authors once again for carefully considering my remarks.

**Limitations:**

Limitations were detailed in the weaknesses section.

**Paper Formatting Concerns:**

None.

**Quality:**

3

**Strengths And Weaknesses:**

### Strengths
The methodology is clear, the results are sound and mathematically well-founded. The contribution is both novel and timely.

### Weaknesses

- Existence of the mean: One key reason why existing approaches for estimating causal effects in the extremes avoid analyzing the Expected Treatment Effect (ETE) is that this quantity does not always exist-unlike quantiles, which are commonly used in such settings. This is why the authors focus on the NETE instead of the ETE. However, beyond the mathematical convenience, it remains unclear what kind of relationships the NETE can capture. In multivariate regular variation, similar standardization techniques are used to quantify dependence. No analogous interpretation is provided in the present work.
- Relevance of the synthetic data experiment: It is not clear from the experiment what specific insights the NETE provides that could not be obtained through a classical causal analysis-assuming the mean exists. The added value of NETE over traditional estimands should be better illustrated or clarified. In partuclar, the claim on l.120-121 is unclear: in general when studying a mean effect, the average, if it exists, it highly impacted by rare events (and not obscured). This point would benefit from further explanations.

---

> ### Author Rebuttal · Authors · 2025-07-31
>
> Thank you for your detailed evaluation of our paper, which will definitely make our paper clearer and better!  Please see the following responses.
>
> 1. Would it be possible to provide deeper insight into what the NETE represents and what type of causal relationship it captures?
>
> Please see the second bullet point of our response to the Reviewer iYgk for a detailed answer.
>
> 2. Similarly, could a synthetic experiment be constructed where the NETE identifies a causal relationship that would not be detected by a classical causal inference method (again, assuming the mean exists)?
>
> In the example in the previous answer, the NETE captures how fast the expected outcome in the tail grows with respect to U. ATE can only capture the change in the overall expectation. While in theory one can estimate CATE and use CATE to capture similar properties, insufficient tailed samples will create a significant challenge for estimation.
>
> 3. Where is the proof for Proposition 3.3? Not sure why this holds.
>
> Thanks for asking. The proof can be found in the appendix, which is in the supplementary material.
>
> ### Clarify questions:
>
> 1. Line 13+: The term “extremities” is somewhat awkward, as it can imply an upper bound. Consider using “extremes” or “extreme events” instead.
>
> Thanks for the suggestion. We will change the wording.
>
> 2. Line 84: The term “consistency” is unclear in this context. Please clarify whether it refers to the consistency of an estimator, or something else.
>
> Thanks for pointing this out. ‘Consistency’ here refers to the consistency assumption in casual inference, which states that an individual's observed outcome is equivalent to their potential outcome under the treatment they actually received.
>
> 3. Lines 94 and 104: Is this the same constant c? If so, it should be explicitly stated; if not, the notation should be differentiated.
>
> Thanks for pointing out the notation abuse. They are different constants. We will explicitly state this in the paper.
>
> 4. Lines 91 and 115: The mean is used before it is formally defined. Consider introducing the necessary definitions earlier for clarity. The orthogonality symbol is used without explanation. Please define this notation when it first appears.
>
> Thanks for pointing these out. We will move the definition of expectation earlier and add the definition of the orthogonality symbol. The orthogonality symbol represents an independent relationship.
>
> 5. Line 94: The term “estimated propensity” is vague—does it refer to an estimated propensity score? If so, please make this explicit.
>
> Thanks for asking. It refers to an estimated propensity score. We will make this explicit.

---

### Official Review · Reviewer_jtyN · 2025-06-29

**Clarity:** 3
**Significance:** 3
**Originality:** 3
**Rating:** 5
**Confidence:** 4

**Summary:**

This paper presents a thorough theoretical analysis of estimating average treatment effects under extreme noise conditions. It adapts tail estimators to the causal inference setting, generalizes identifiability theorems under commonly assumed conditions, and supports its results with experiments on synthetic (and semi-synthetic) data.

**Questions:**

I added the questions in the weaknesses above, please read above for further details.
Three things that, in my opinion, would significantly increase the value of the paper:

- A more thorough literature review
- Clearer mathematical statements (it was sometimes difficult to identify the definition of each symbol)
- Experiments: discussion of the effect of missing confounders on stability, explanation of stability with respect to the number of samples, and inclusion of error bars (especially as a function of the number of samples, to test the hypothesis mentioned by the authors)

**Ethical Concerns:**

["NO or VERY MINOR ethics concerns only"]

**Final Justification:**

The paper is technically solid and it studies a new interesting problem. If the authors add a more complete literature review and make the edits that are included in their response I think the paper is good shape. I will summarize the issues that are resolved (if they ll be added to the revised manuscript):

Summary of issues resolved:
- more complete literature review.
- adding error bars.
- clarification about some of the mathematical statements.

**Limitations:**

Yes. I don't think any important limitation is missing as the authors state the assumptions under which their analysis hold.

**Paper Formatting Concerns:**

I do not see any formatting issues.

**Quality:**

3

**Strengths And Weaknesses:**

### Strengths

- I appreciate the motivation to estimate treatment effects under extreme scenarios which makes the scenario studied by the paper relevant and unique to the best of my knowledge.
- The theoretical analysis is thorough.

### Weaknesses

- Since there is no widely agreed-upon definition of extreme treatment effects in the literature, and the authors appear to propose a new one, it would be helpful to state this more clearly in the abstract (e.g., "introducing a new definition to model extremities in treatment effects").

- The sentence *"Classic causal inference literature mainly focuses on estimating the average effects among certain groups"* gives the impression that no prior work has modeled quantities beyond the population average (ATE). However, there is a substantial body of work on estimating conditional average treatment effects (CATE) [1,2], modeling distributions [3,4], and on tail/extreme treatment effects [5,6,7].

- As previously mentioned, a quick search reveals at least two papers available online that tackle the intersection of Extreme Value Theory (EVT) and causal inference in similar (though not identical) settings — namely, Huang et al. (2022) and Aloui et al. (2023). A more thorough literature review focusing on this intersection is necessary. Both papers cite additional related works. While the current paper focuses on the average treatment effect under heavy-tailed exogenous variables — a different setting — the use of EVT theory is highly relevant and similar. A discussion of these differences would strengthen the literature review.

- The authors only study an average (aggregate) quantity, not one conditioned on covariates. I believe conditioning on covariates would make the contribution more complete and relevant. For example, in healthcare, we may care about the extreme response of a treatment for a specific individual, such as

 $$\theta^{\text{NETE}}(X) = \lim_{t \rightarrow \infty} \mathbb{E}\left[\frac{Y(1) - Y(0)}{t^{\alpha}} \middle|\ X, |U| > t\right]$$

Do the authors see a natural extension of their work to this conditional setting?

- In Assumption 3.2, I find the statement is unclear. Should it state that there exists some $\alpha$ ? Is $\alpha$ unique for a fixed $g$? Also, is $e(t)$ a function of $g$? That is, should the statement read: `` $\exists g, \exists e$ such that …”? Clarification is needed for me to understand the statement better, and $\alpha$ should be more explicitly defined. I recall seeing similar assumptions in EVT literature — could the authors point to any related work? Also, does the property hold only as $t \to \infty$, or for all $t$?

- Regarding the statement *“E[Y(d)] is of the order $O(\|U\|^{\alpha})$ and (3.2)”* — does this mean that $\alpha$ is fixed for each $Y$? Please clarify.

- The sentence *“In these cases, $f$ grows polynomially with respect to $\|U\|$ and $e(t) = 0$ exactly.”* refers to just one choice of \(e\). However, many functions \(e(t)\) could verify the required property, even if they are nonzero but converge to zero as $t \to \infty$. Is my understanding correct?

- The claim that the definition *“generalizes ATE to the setting of extreme events and aligns with the growth rate given by EVT”* is somewhat misleading. It is not obvious how this is a generalization of ATE. Rather, it appears to be a separate definition. It may be more accurate to say it generalizes the *unnormalized* version (ETE), if anything. Or am I missing something?

- Could the authors add error bars in Figure 2? The experiments are repeated over 50 different seeds, meaning the data changes each time, and reporting variance would be helpful.

- The authors mention that as the number of samples increases, the assumption becomes violated and the estimator becomes dominated by error. This is quite counterintuitive: if the estimator is consistent, it should converge to the true estimand as the sample size grows. Could the authors elaborate on this phenomenon? Is there a way to filter out “bad samples” that lead to this behavior?

- A final question: Could the authors expand on how they envision extending this work to the estimation of conditional average treatment effects? Also, what do they expect regarding the performance of their estimator when conditional unconfoundedness is violated? While this question likely deserves a dedicated study, if the authors have time during the rebuttal phase, it would be informative to include a synthetic experiment where $X \sim \text{Unif}[0,1]^d$ with $d$ ranging from 5 to 10, and only the first 5 dimensions used to adjust for bias (as is done now). How would the performance degrade as dimensionality increases (i.e., as more confounders are missing)? While not a formally defined sensitivity analysis, this could still help us understand the robustness of the method to unobserved confounding.


#### References

[1] Shalit, Uri, Fredrik D. Johansson, and David Sontag. "Estimating individual treatment effect: generalization bounds and algorithms." International conference on machine learning. PMLR, 2017.

[2] Abrevaya, Jason, Yu-Chin Hsu, and Robert P. Lieli. "Estimating conditional average treatment effects." Journal of Business & Economic Statistics 33.4 (2015): 485-505.

[3] Gautier, Eric, and S. T. E. F. A. N. Hoderlein. "Estimating the distribution of treatment effects." arXiv preprint arXiv:1109.0362 (2013).

[4] Hohberg, Maike, Peter Pütz, and Thomas Kneib. "Treatment effects beyond the mean using distributional regression: Methods and guidance." PloS one 15.2 (2020): e0226514.

[5] Huang, Wei, Shuo Li, and Liuhua Peng. "Extreme continuous treatment effects: Measures, estimation and inference." arXiv preprint arXiv:2209.00246 (2022).

[6] Aloui, Ahmed, et al. "Treatment Effects in Extreme Regimes." arXiv preprint arXiv:2306.11697 (2023).

[7] Bodik, Juraj. "Extreme treatment effect: Extrapolating dose-response function into extreme treatment domain." Mathematics 12.10 (2024): 1556.

---

> ### Author Rebuttal · Authors · 2025-07-31
>
> Thanks for the detailed feedback and valuable suggestions! Please see the following responses.
>
> 1. It would be helpful to state clearly in the abstract that the authors appear to propose a new definition of extreme treatment effects.
>
> Thanks for the advice. We will add a sentence in our abstract to make this clearer.
>
> 2. The paper gives the impression that no prior work has modeled quantities beyond the population average (ATE). A discussion of these differences would strengthen the literature review.
>
> Thanks for providing the related reference. We acknowledge that the expression of the sentence may be inappropriate. We will provide a more thorough literature review in later versions. Currently, there are two approaches in the literature to characterize the extreme treatment effect (ETE). The first approach is to use extreme Quantile Treatment Effect (QTE) [3, 4, 5, 6], which we discuss in detail in our paper. The other approach is to assume the outcome falls in the domain of attraction and use the change of Extreme Value Index (EVI) as a measure for ETE [1, 2]. In particular, [3] also considers using the ratio of tail expectation under different treatments as a measure. However, our setting is different from these works in that (1) We allow extreme exogenous variables and the outcome is a function of treatment, covariate and the extreme variables. (2) We model extreme events by a multivariate regularly varying variable, while all these works focus on univariate settings. Moreover, our work can also be placed in the broader context in the literature that studies causal effects of treatment beyond the mean, for example, distributional effect [7,8,9], Ratio‐ and Log‐Ratio‐Based Effects [10,11], CATE [12,13,14].
>
> [1] Aloui, Ahmed, et al. "Treatment Effects in Extreme Regimes." arXiv preprint arXiv:2306.11697 (2023).
>
> [2] Bodik, Juraj. "Extreme treatment effect: Extrapolating dose-response function into extreme treatment domain." Mathematics 12.10 (2024): 1556.
>
> [3] Huang, Wei, Shuo Li, and Liuhua Peng. "Extreme continuous treatment effects: Measures, estimation and inference." arXiv preprint arXiv:2209.00246 (2022).
>
> [4] Victor Chernozhukov and Songzi Du. Extremal Quantiles and Value-at-Risk, May 2006.
>
> [5] Yichong Zhang. Extremal quantile treatment effects. The Annals of Statistics, 46(6B):3707–3740,December 2018.
>
> [6] David Deuber, Jinzhou Li, Sebastian Engelke, and Marloes H. Maathuis. Estimation and Inference of Extremal Quantile Treatment Effects for Heavy-Tailed Distributions. Journal of the American Statistical Association, 119(547):2206–2216, July 2024.
>
> [7] Abadie, A. (2002). Bootstrap Tests for Distributional Treatment Effects in Instrumental Variable Models. J. Amer. Statist. Assoc., 97(457): 284–292.
>
> [8] Gautier, Eric, and S. T. E. F. A. N. Hoderlein. "Estimating the distribution of treatment effects." arXiv preprint arXiv:1109.0362 (2013).
>
> [9] Hohberg, Maike, Peter Pütz, and Thomas Kneib. "Treatment effects beyond the mean using distributional regression: Methods and guidance." PloS one 15.2 (2020): e0226514.
>
> [10] Cole, S. & Hernán, M. (2002). Fallibility in Estimating Direct Effects. Epidemiology, 13(3): 297–306.
>
> [11] VanderWeele, T.J. (2014). Policy‐Relevant Proportions for Direct Effects. Epidemiology, 24(2): 175–176.
>
> [12] Shalit, U., Johansson, F. & Sontag, D. (2017). Estimating individual treatment effect: generalization bounds and algorithms. International Conference on Machine Learning (ICML).
>
> [13] Wager, S. & Athey, S. (2018). Estimation and inference of heterogeneous treatment effects using random forests. Journal of the American Statistical Association, 113(523), 1228–1242.
>
> [14] Abrevaya, Jason, Yu-Chin Hsu, and Robert P. Lieli. "Estimating conditional average treatment effects." Journal of Business & Economic Statistics 33.4 (2015): 485-505.
>
>
>
> 3. Do the authors see a natural extension of their work to this conditional setting?
>
> Thanks for the great question. Indeed, our results can be extended to the conditional setting using a similar argument without much difficulty. The identification formula (Proposition 3.3) can be adapted to
> $$ \theta^{{CNETE}}(x) = \lim_{t \rightarrow \infty} \mathbb{E}[g (X, 1, U/\|U\|) - g (X, 0, U /\|U\|) \mid \|U\|> t, X = x] \cdot \lim_{t \rightarrow \infty} \mathbb{E}[\|U\|^{\alpha} / t^{\alpha} \mid \|U\|> t] . $$
> Similar to CATE estimation, one can design different estimators based on this identification formula, e.g., T learner, S learner, X learner, DR learner.
>
> 4. Clarification of  Assumption 3.2.
>
> Thanks for the question and sorry for the confusion. The correct statement should be that there exists some constant $\alpha$, functions $g(x,d,u), e(t)$ such that the inequality holds for all x, t, u, d. It is easy to see that $ \lim_{t\rightarrow \infty} f(x,d,tu)/t^\alpha = g(x,d,u)$. Therefore, the constant $\alpha$ is unique. Otherwise, if there exists $\alpha’$ that satisfies the same relationship, we have $\lim_{t\rightarrow \infty} t^{|\alpha - \alpha’|} <\infty$. Since $\alpha$ is unique, by this relationship, $g$ is unique too. The error term $e(t)$ is not a function of $g$ and only depends on t. This assumption can be viewed as an asymptotic version of homogeneous functions.
>
> 5. Is $\alpha$ fixed for each $Y$? Please clarify.
>
> Thanks for the question. Yes. $\alpha$ is fixed for each $Y(d)$. Furthermore, for different $d$, the alpha is the same.
>
> 6. Many functions $e(t)$ could verify the required property, even if they are nonzero but converge to zero as $t\rightarrow \infty$. Is my understanding correct?
>
> Yes! Your understanding is correct. Since $e(t)$ is only an upper bound, it can be satisfied by many functions as long as they converge to 0 as t grows to infinity.
>
> 7. The claim that the definition “generalizes ATE to the setting of extreme events and aligns with the growth rate given by EVT” is somewhat misleading. It is not obvious how this is a generalization of ATE. Rather, it appears to be a separate definition. It may be more accurate to say it generalizes the unnormalized version (ETE), if anything. Or am I missing something?
>
> Thanks for the suggestion! We intended to say that NETE provides a causal measure in the extreme setting just like ATE in the regular setting. Since this expression may cause confusion, we will change it in the later version.
>
> 8. Could the authors add error bars in Figure 2? The experiments are repeated over 50 different seeds, meaning the data changes each time, and reporting variance would be helpful.
>
> Thanks for pointing that out. We will include error bars in our figures in later versions.  Unfortunately, we did not find a place to show our figures here.
>
> 9. The authors mention that as the number of samples increases, the assumption becomes violated and the estimator becomes dominated by error.
>
> Thanks for the question. We want to clarify that in the second experiment (Fig 2 in the paper), we design the data generation process such that assumption 3.4 (Pareto mixture) is violated all the time. Since our non-asymptotic analysis is based on this assumption, violation of this assumption may cause the error bound in Theorem 3.5 to be invalid. We will make this point clearer in our experiments section. There is a way to detect whether assumption 3.4 is violated. Please see the third point of our response to Reviewer GEJF for explanation.
>
> 10. What do authors expect regarding the performance of their estimator when conditional unconfoundedness is violated?
>
> Thanks for the questions. When the conditional unconfoundness assumption is violated, our identification formula no longer holds, and thus our estimator is non-consistent. There have been many works in the literature discussing this assumption. One approach is to completely drop this assumption and consider using partial identification methods to obtain a bound for the estimand [1,2]. The other approach is to conduct a sensitivity analysis [3]. Following the reviewer’s advice, we examine what would happen empirically if this assumption is violated. Our data generation process is
>
> \begin{align*}
> X \sim \mathrm{Uniform}([0,1]^{10}),\  p_i = \frac{1}{1 + \exp\bigl(-\beta^\top (X \circ X)\bigr)}, \quad D_i \sim \mathrm{Bernoulli}(p_i), \quad
> \end{align*}
> and
> $$Y = \|U\|^2(D + \gamma^T(X \circ X) + U/\|U\| + \epsilon) + \|U\|,$$
> where $U,\epsilon$ are generated the same way as in the first synthetic experiment in the paper and $X \circ X$ is the element-wise multiplication. The following table shows that the MSE decreases as the observable dimension increases.
>
> | # of Dimensions | EVT-DR MSE |
> |----------------:|-----------:|
> |               5 |      0.2016 |
> |               6 |      0.1145 |
> |               7 |      0.0652 |
> |               8 |      0.0419 |
> |               9 |      0.0204 |
> |              10 |      0.0133 |

---

> > ### Comment · Reviewer_jtyN · 2025-08-01
> >
> > Thank you for your detailed response! My questions have been addressed. I highly recommend incorporating the edits included in the rebuttal in the revised version. I would have loved to see a more complete study of the conditional setting in the main paper as it seems for me the more interesting case from a machine learning perspective. That said, my evaluation of the paper remains very positive and I have a better understanding of it after the rebuttal

---

### Official Review · Reviewer_GEJF · 2025-07-02

**Clarity:** 2
**Significance:** 3
**Originality:** 3
**Rating:** 4
**Confidence:** 4

**Summary:**

The work introduces Normalized Extreme Treatment Effect (NETE), a causal estimand that targets how an intervention shifts tail outcomes—e.g. damage during 100‑year storms—rather than means. Building on multivariate regular variation, the authors:

- formalise NETE and prove identifiability under potential‑outcome exogeneity plus an “asymptotic homogeneity” growth condition

- derive a decomposition that separates spectral‑measure and α‑moment terms, enabling plug‑in estimation

- craft two estimators—IPW and doubly‑robust (DR)—combined with an adaptive Hill tail‑index estimator

- provide non‑asymptotic error bounds under a structured Pareto‑mixture model

- validate the approach on extensive synthetic grids and a wave‑surge semi‑synthetic dataset, where EVT‑DR consistently attains the lowest MSE versus naïve baselines

**Questions:**

Please see the section on strengths and weaknesses.

**Ethical Concerns:**

["NO or VERY MINOR ethics concerns only"]

**Limitations:**

Limitations have not been discussed.

**Paper Formatting Concerns:**

No formatting concerns.

**Quality:**

3

**Strengths And Weaknesses:**

### Strengths
- Novel Problem Framing. NETE fills a genuine gap by marrying EVT with causal inference to quantify tail counterfactuals—distinct from existing QTE work.

- Methodological Rigor. Doubly‑robust construction and adaptive Hill estimation yield consistency plus finite‑sample rates.

- Synthetic Thoroughness. Four‑way sweeps over tail index, dimension, and growth exponent highlight estimator robustness.

- Clear Mathematical Exposition. Assumptions and steps are easy to trace; proofs provided.

### Weaknesses & Questions for the Authors
1. Observability of Extreme Noise U. Real‑world practitioners seldom measure the latent driving shock; instead Y alone reflects severity. How realistic is it to assume ∥U∥ is observed so that {∥U∥ > t} can be conditioned on? Could the method be adapted when only Y (or a proxy) identifies extremes?

2. Independence Assumption. Assumption 2.1 treats U independent of treatment assignment after conditioning on X. In many policy settings the exposure and hazard covary (e.g. flood defences built where storms frequent). Please discuss robustness / sensitivity when this fails.

3. Tail‑Model Specification. Non‑asymptotic bound (Thm 3.5) hinges on Pareto‑mixture Assumption 3.4; yet Fig. 2 shows violation degrades MSE. Can the authors provide a diagnostic or adaptive procedure to detect mis‑specification in practice?

4. Scaling‑Exponent α. α is estimated by a log‑log regression; no theory is provided for its error propagation. Empirically, how sensitive is NETE to mis‑estimated α?

5. Compute & Reproducibility. Please provide (i) code, (ii) random seeds, (iii) hardware spec, and (optionally) (iv) wall‑clock times so reviewers can replicate Tables 1–2.

6. Real‑world Case Study. A genuine observational dataset—e.g., hurricane damage with/without building codes—would greatly strengthen practical relevance.

7. Why are only averages shown in Fig 2 although the experiment was repeated 50 times. Please add CI.

8. A paragraph on Limitations is missing.

---

> ### Author Rebuttal · Authors · 2025-07-31
>
> Thank you for the valuable feedback! Please see the following detailed response to your questions. We hope our responses can resolve your concern.
>
> 1. Observability of Extreme Noise U. How realistic is it to assume ∥U∥ is observed so that {∥U∥ > t} can be conditioned on? Could the method be adapted when only Y (or a proxy) identifies extremes?
>
> Thanks for the great question. We can think of U as some factors that model the extreme events, e.g, speed and radius of a hurricane, which are often observable. In many cases, we are interested in the influence of extreme events rather than the events themselves. For example, we care about how much economic loss would be caused by the extreme events. That’s why in our model, the outcome is a function of U, X (covariate) and D (treatment). We will make this point clear in our paper. When only Y identifies the extremes, the identification formula (Proposition 3.3) no longer holds and we need to develop new identification techniques and change our framework. Indeed, in the literature, [1,2,3] consider such settings and develop estimation methods for their proposed estimands.
>
> [1] Aloui, Ahmed, et al. "Treatment Effects in Extreme Regimes." arXiv preprint arXiv:2306.11697 (2023).
>
> [2] Bodik, Juraj. "Extreme treatment effect: Extrapolating dose-response function into extreme treatment domain." Mathematics 12.10 (2024): 1556.
>
> [3] Huang, Wei, Shuo Li, and Liuhua Peng. "Extreme continuous treatment effects: Measures, estimation and inference." arXiv preprint arXiv:2209.00246 (2022).
>
> 2. Independence Assumption. Please discuss robustness/sensitivity when this fails.
>
> Thanks for the valuable feedback. We acknowledge that Assumption 2.1 may not always hold in practice, as policies are often influenced by the extreme event U. Regarding the situations where the independence assumption is violated, there are two cases. First, we have prior knowledge about which extreme variables influence the treatment assignment. In this case, we suggest including the extreme variables as covariates. Please see our response to the first question to Reviewer iYgk for detailed discussion. Second, we do not know which extreme variables will influence the treatment, but we treat U to be independent of treatment. In this case, the IPW estimator will be biased since the propensity estimation does not include (part of) U as inputs and the estimated propensity function will not converge to the ground-truth propensity. However, if all other assumptions in Theorem 3.5 still hold, the DR estimator is still consistent thanks to the double robust property.
>
> 3. Tail‑Model Specification. Can the authors provide a diagnostic or adaptive procedure to detect mis‑specification in practice?
>
> Thanks for the important question. Unfortunately, in theory, we don’t have a good way to test whether Assumption 3.4 holds or not. In practice, we suggest plotting the spectral distribution $U/ \|U\| \mid \|U\| > t$ for a large threshold t. If Assumption 3.4 holds, by Lemma 2.1 in [1], we should be able to see points concentrate on a few clusters. If points spread evenly on the sphere, Assumption 3.4 may be violated.
>
> [1] Zhang, Xuhui, et al. "Wasserstein-based minimax estimation of dependence in multivariate regularly varying extremes." arXiv preprint arXiv:2312.09862 (2023).
>
> 4. Scaling‑Exponent α. α is estimated by a log‑log regression; no theory is provided for its error propagation. Empirically, how sensitive is NETE to misestimated α?
>
> This is a great question! We did an experiment to see the sensitivity of alpha. In particular, we compare the MSE of using the true $\alpha$ versus using our heuristic to estimate the $\alpha$ in the first synthetic experiment in our paper. Unfortunately, we did not find a place to show our figures here. The following table shows part of the results. We can see that actually the MSEs are closed and our heuristic for $\alpha$ estimation works well.
>
> | Sample size | MSE (est α) | SE (est α) | MSE (true α) | SE (true α) |
> |-----------:|------------:|-----------:|-------------:|-----------:|
> |       1,000 |      1.4982 |    1.0741 |       1.2913 |    0.8634 |
> |       2,000 |      0.6739 |    0.6832 |       0.5383 |    0.5219 |
> |       3,000 |      0.3911 |    0.3724 |       0.5100 |    0.3942 |
> |       4,000 |      0.3333 |    0.3664 |       0.3625 |    0.3329 |
> |       5,000 |      0.2499 |    0.2726 |       0.3246 |    0.3023 |
>
> 5. Compute & Reproducibility. Please provide (i) code, (ii) random seeds, (iii) hardware spec, and (optionally) (iv) wall‑clock times so reviewers can replicate Tables 1–2.
>
> Thanks for asking. We will provide the code in the camera-ready version. Unfortunately, due to Neurips' policy, we cannot provide any link in the rebuttal stage.
>
> 6. Real‑world Case Study. A genuine observational dataset—e.g., hurricane damage with/without building codes—would greatly strengthen practical relevance.
>
> Thanks for the suggestion. We want to mention that it is difficult to find realistic datasets that include covariates, extreme events and outcomes. To conduct such a real-world study, we need to collect data from several different data sources and construct a new dataset, which requires a substantial amount of time and effort. Given that the goal of this paper is to propose a new framework for estimating extreme effects, we think that synthetic and semi-synthetic experiments are enough to verify our theoretical results. We will work on real-world problems in later studies.
>
> 7. Why are only averages shown in Fig 2 although the experiment was repeated 50 times. Please add CI.
>
> Thanks for pointing that out. We will add the error bar to all graphs. Unfortunately, we did not find a place to show our figures here.
>
> 8.  A paragraph on Limitations is missing.
>
> Thanks for the feedback. We briefly mention the limitation in the conclusion section in the appendix. We will add a more detailed explanation of the limitation.

---

### Decision · Program_Chairs · 2025-09-17

**Decision:**

Accept (poster)

**Comment:**

The standard causal inference methods do not work for the data with extreme values. In this paper, the authors propose a method for treatment effects estimation in extreme data to capture the causal effect at the occurrence of rare events of interest. They define Normalized Extreme-value Treatment Effect (NETE) and use the theory of multivariate regular variation to model extremities. They also propose a doubly robust estimation method.

The work has its limitations: some assumptions such that observability of the extreme noise, independence Assumption  2.1, Pareto‑mixture Assumption 3.4 are limiting but do not render the methodology ineffective. The literature review can also be improved. Please address the rest of the reviewers' feedback in the final version of the paper.